# A global lipid map defines a network essential for Zika virus replication

Hans C. Leier [1], Jules B. Weinstein [1], Jennifer E. Kyle[2], Joon-Yong Lee [2], Lisa M. Bramer [3], Kelly G. Stratton[3], Douglas Kempthorne[1,4], Aaron R. Navratil[5], Endale G. Tafesse[6], Thorsten Hornemann [7], William B. Messer[1,8], Edward A. Dennis [5], Thomas O. Metz [2], Eric Barklis[1] & Fikadu G. Tafesse [1✉]

Zika virus (ZIKV), an arbovirus of global concern, remodels intracellular membranes to form replication sites. How ZIKV dysregulates lipid networks to allow this, and consequences for disease, is poorly understood. Here, we perform comprehensive lipidomics to create a lipid network map during ZIKV infection. We find that ZIKV significantly alters host lipid composition, with the most striking changes seen within subclasses of sphingolipids. Ectopic expression of ZIKV NS4B protein results in similar changes, demonstrating a role for NS4B in modulating sphingolipid pathways. Disruption of sphingolipid biosynthesis in various cell types, including human neural progenitor cells, blocks ZIKV infection. Additionally, the sphingolipid ceramide redistributes to ZIKV replication sites, and increasing ceramide levels by multiple pathways sensitizes cells to ZIKV infection. Thus, we identify a sphingolipid metabolic network with a critical role in ZIKV replication and show that ceramide flux is a key mediator of ZIKV infection.

[1] Department of Molecular Microbiology & Immunology, Oregon Health & Science University (OHSU), Portland, OR 97239, USA. [2] Biological Sciences Division, Earth and Biological Sciences Directorate, Pacific Northwest National Laboratory (PNNL), Richland, WA 99352, USA. [3] Computing and Analytics Division, National Security Directorate, PNNL, Richland, WA 99352, USA. [4] Center for Diversity and Inclusion, OHSU, Portland, OR 97239, USA. [5] Departments of Chemistry & Biochemistry and Pharmacology, University of California San Diego School of Medicine, La Jolla, CA 92093, USA. [6] Department of Plant Sciences, College of Agriculture and Bioresources, University of Saskatchewan, Saskatoon, SK S7N 5A8, Canada. [7] University Zurich and University Hospital Zurich, University of Zurich, Zurich 8091, Switzerland. [8] Department of Medicine, Division of Infectious Diseases, OHSU, Portland, Oregon 97239, USA. ✉email: tafesse@ohsu.edu

Zika virus (ZIKV), an enveloped positive-strand RNA virus of the family Flaviviridae, has recently emerged as a significant global human health threat[1]. Following its rapid expansion into the Americas, ZIKV has been found to possess a unique combination of virulence traits, including the ability to cross the human placental barrier and cause microcephaly and other congenital abnormalities[2]. Like other positive-strand RNA viruses, ZIKV is highly dependent on host cell machinery for the production of new virions[3,4]. The resulting disruption of cellular networks can directly contribute to clinical disease, raising the need for system biology methods to elucidate host–virus interactions within and between all host tissue compartments[5,6].

The development of genome-scale CRISPR/Cas9 knockout screens has tremendously advanced investigations into genetic factors that affect disease[7]. Knockout screens for genes required for flavivirus-induced cell death have identified a number of novel targets, including multiple components of the endoplasmic reticulum (ER) protein processing and quality-control pathways[8,9]. While such screens are powerful tools for elucidating host factors in infection, they have important limitations: knockouts in genes that are not essential for virus-induced cell death may escape detection, as may knockouts in host pathways with genetic or functional redundancy that can continue to function in the absence of individual gene products. To date, host factors identified in gene-editing experiments have thus far clustered around ER protein complexes essential for flavivirus genome replication and translation, while metabolic pathways with important roles during infection have been mostly underrepresented.

ZIKV's distinct tropism and pathology, and lack of readily apparent disease mechanism, has prompted efforts to systematically map the interaction of ZIKV infection with host cells[10]. While studies of the ZIKV-infected host proteome[3], transcriptome[4,9,11], and protein–protein interactome[12] have yielded new insights into ZIKV biology, the mechanistic basis of ZIKV pathogenesis remains largely unknown.

Like other flaviviruses, ZIKV carries out each stage of its replication cycle in close association with cellular membranes, including the synthesis of new genome copies and assembly of viral particles within specialized replication complexes (RCs) formed from extensively remodeled ER membranes[13]. These steps appear to require a specific lipid milieu, as flaviviruses presumably modify various host lipid pathways to create this milieu[14–16]. A rapidly growing body of knowledge on the importance of lipids in cell organization, signaling networks, and viral disease outcomes therefore led us to investigate how ZIKV perturbs cellular lipid metabolic networks to establish and promote infection[17,18].

To systematically map the host lipid–virus interaction networks in an unbiased manner, we have carried out a global lipidomic survey in human cells infected with ZIKV or ectopically expressing NS4B, one of the nonstructural proteins of ZIKV known to be involved in forming viral replication sites[19]. We show that ZIKV infection as well as NS4B expression significantly alters the lipid composition of human cells, with the most striking pattern of changes seen within sphingolipids. Ceramide, a bioactive sphingolipid implicated in signaling and apoptosis, is recruited to ZIKV replication sites and strongly associates with ZIKV NS4B. We use pharmacological inhibition and genetic knockouts of enzymes involved in sphingolipid biosynthesis in various cell types, including neural progenitor cells, to demonstrate that sphingolipids are required for ZIKV replication but not for binding to or entering host cells. Conversely, genetic knockout of sphingomyelin synthesis drastically increases cellular permissiveness to ZIKV, indicating that ceramide or its derivatives, rather than sphingomyelin, are required for ZIKV infection.

Together, our study identifies a sphingolipid metabolic network with a novel proviral role in ZIKV replication.

## Results

**ZIKV alters the lipid landscape of host cells**. To understand how cellular lipid metabolism is altered by ZIKV infection, we carried out a global lipidomic survey of the model Huh7 human hepatic carcinoma cell line[20] infected for 24 or 48 h with Asian lineage ZIKV strain FSS13025 (Fig. 1a). Lipids extracted from populations of mock and infected cells ($n = 5$ biological replicates per condition) were analyzed with electrospray ionization tandem mass spectrometry (LC–ESI–MS/MS) (Supplementary Data 1 and Supplementary Fig. 1a–f). We identified 340 lipid species spanning the phospholipid, sphingolipid, glycerolipid, and sterol classes for which pairwise comparisons of normalized abundance between mock and infected cells could be made (Supplementary Data 1). Of these, 80 species (23.5%) showed significant changes in abundance by 24 h post infection (hpi) and 172 species (50.6%) were significantly altered by 48 hpi ($P < 0.05$, analysis of variance (ANOVA) or g test) (Supplementary Data 1). Principal component analysis (PCA) of these observations confirmed that infection status (mock or ZIKV) and timepoint (24 or 48 hpi) accounted for most of these changes, with changes in lipid composition between mock and infected cells increasing over time (Fig. 1b).

Next, we examined how ZIKV-induced changes in host lipid composition broke down by subclass and species (Fig. 1c, d). A map of the pairwise correlations of all 340 species at 48 hpi (Supplementary Fig. 2a) revealed that lipid subclasses largely fell into two groups of species that were either enriched or depleted in abundance (Supplementary Fig. 2b), suggesting that individual metabolic pathways are up- or downregulated to create a specific lipid milieu around the events of the viral replication cycle. Supporting this, many of the trends we observed were consistent with earlier reports of functional roles for lipids during flavivirus infection. In line with evidence that lipid droplets are consumed as an energy source during flavivirus replication, most triglycerides (TG) declined over the course of infection, though TG species with 22:6 acyl chains increased. All cholesterol esters were enriched in ZIKV-infected cells, reproducing trends seen during dengue virus infection. Trends among phospholipid subclasses varied: cardiolipin, phosphatidylserine (PS), and phosphatidylethanolamine species were mostly depleted at 24 and 48 hpi, and phosphatidylcholine species were enriched. A notable exception was the phosphatidylinositol (PI) subclass, which went from largely depleted to largely enriched between 24 and 48 hpi. The role of PI signaling in regulating numerous cellular functions is well established, and our data support findings that PI pathways are upregulated to block apoptosis late in flavivirus infection.

**Expression of ZIKV NS4B enriches host sphingolipids**. The flavivirus genome encodes three structural (capsid [C], envelope [E], and membrane [prM]) and seven nonstructural (NS) proteins (NS1, NS2A, NS2B, NS3, NS4A, NS4B, and NS5). Structural ZIKV proteins carry out the entry and membrane fusion steps of the viral life cycle[21], while NS proteins cooperatively remodel ER membranes to form replication sites and synthesize viral RNA[22]. Despite their limited size and number, the functions of most of the NS proteins are poorly characterized[23], as are their interactions with host lipids[24] and potentially hundreds of unique proteins[3,12,25]. While the enigmatic nature of the ZIKV NS proteins and their interactions presented challenges to defining a mechanistic basis for our lipidomics results, two lines of evidence led us to investigate NS4B as potentially important in altering lipid metabolism. First, NS4B is a transmembrane protein that

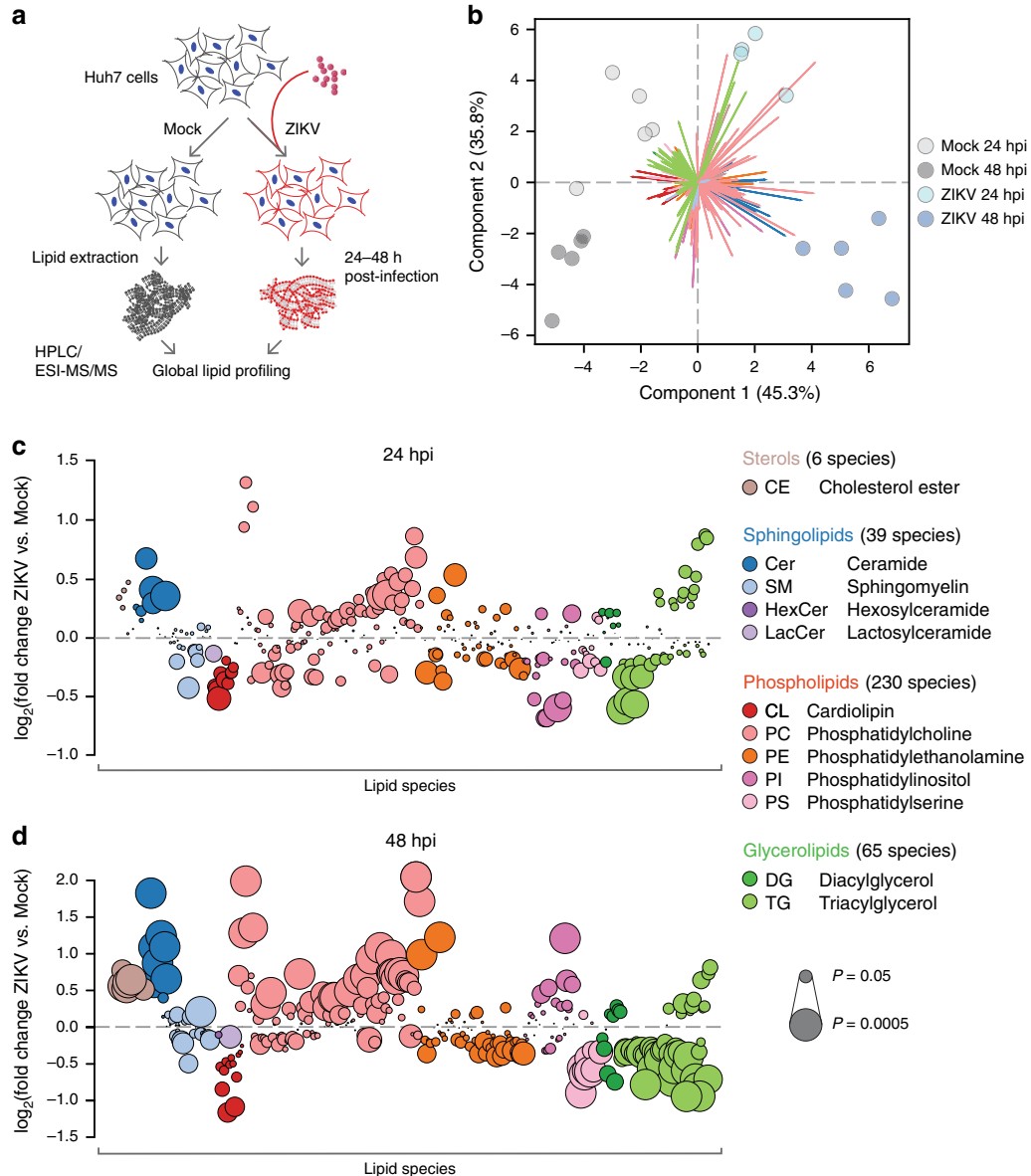

**Fig. 1 Global lipidomics of ZIKV-infected human cells. a** Study overview. Huh7 cells were infected with ZIKV strain FSS13025 for 24 or 48 h. Each experimental condition (Mock 24 hpi, ZIKV 24 hpi, Mock 48 hpi, and ZIKV 48 hpi) had $n = 5$ replicates for a total of 20 biological samples, 19 of which were included in our final analysis. **b** Principal component analysis (PCA) of the lipidomics dataset. Colored arrows represent individual lipid species. **c**, **d** Bubble plots of log2 fold changes in abundance of lipid species in ZIKV-infected cells relative to mock at 24 hpi (**c**) and 48 hpi (**d**). Bubble size represents $P$ value from one-way ANOVA or g test. See also Supplementary Fig. 1, Supplementary Data 1, and the Source Data file.

produces the strongest ER stress and autophagic response of the ten flavivirus proteins when individually expressed[26,27], and lipid metabolism is coordinately regulated with these pathways during periods of stress[28–31]. Second, the NS4B of the closely related *Flaviviridae* member Hepatitis C virus (HCV) dysregulates lipid metabolism to permit viral replication[32], which may directly contribute to liver disease[33]. Like *Flavivirus* NS4B[34], HCV NS4B is an integral component of the viral RC, and can both remodel ER membranes into replication site-like structures[35] and induce a potent ER stress response[36] when individually expressed.

To examine whether ZIKV NS4B could similarly regulate global lipid metabolism, we performed a second lipidomic survey of HEK 293T cells transfected with ZIKV NS4B-FLAG or an empty vector control (Fig. 2a, Supplementary Fig. 3a–d). Supporting its role as a major factor in host–virus interactions, NS4B caused significant downregulation or upregulation ($P <$

0.05, one-way ANOVA) in 44% of the 318 lipid species identified relative to the control (Fig. 2b and Supplementary Data 2). Furthermore, many of these changes were comparable to or exceeded those of similar species in ZIKV-infected cells, especially in the negative direction (Fig. 2c).

We analyzed the relationship between NS4B transfection and ZIKV infection for the set of lipid species ($n = 98$) that appeared in both datasets, and found a weak but highly statistically significant positive correlation between the two conditions ($P < 0.05$, Pearson correlation coefficient) (Fig. 2d). Strikingly, ceramides were the only species that were significantly enriched by over a log in both conditions (Fig. 2d). When we repeated our analysis for individual lipid classes, sphingolipids were even more strongly correlated than total lipids, or phospholipids and glycerolipids alone (Fig. 2e). Together, our results support a causal relationship between NS4B expression and targeted

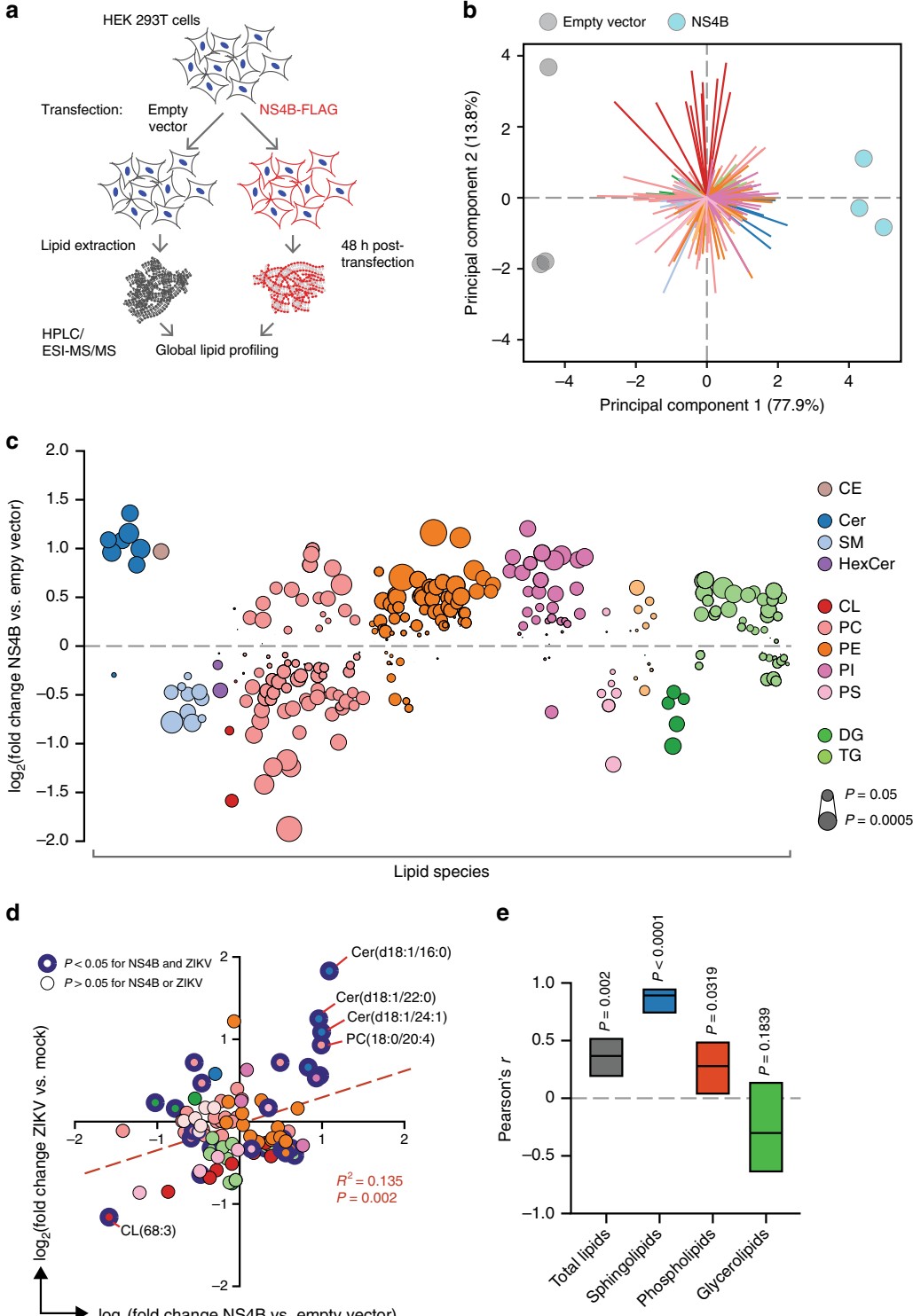

**Fig. 2 ZIKV NS4B dysregulates host lipid metabolism. a** Design of transfection experiment. Total lipids were extracted from HEK 293T cells transfected with NS4B or an empty vector control; $n = 3$ biological replicates per condition. **b** PCA of the lipidomics dataset. Colored arrows represent individual lipid species. **c** Bubble plots of log2 fold changes in abundance of lipid species in ZIKV-infected cells relative to mock at 24 hpi. Bubble size represents $P$ value from one-way ANOVA or g test. **d** Correlation of log2 fold-change values of lipid species ($n = 95$) identified in both ZIKV-infection and NS4B-transfection experiments (see Supplementary Data 1 and 2, respectively). Linear regression best-fit line ($y = 0.3212 \times 0.02533$), $R^2$, and $P$ values are shown. **e** Pearson's correlation coefficient ($r$) for total or subclasses of lipid pairs. Lines and bars are $r$ values with 95% CI, respectively. See also Supplementary Fig. 3, Supplementary Data 2, and the Source Data file.

regulation of sphingolipid metabolism, along with previously described stress-signaling pathways.

**Depletion of cellular sphingolipids inhibits ZIKV propagation.** De novo sphingolipid biosynthesis begins in the ER with the condensation of L-serine and palmitoyl CoA catalyzed by serine palmitoyltransferase (SPT) (Fig. 3a). Further reactions yield dihydrosphingosine, which is converted by ceramide synthase (CerS) to dihydroceramide, then to ceramide (Cer), the precursor to SM and other downstream sphingolipids. To determine if ZIKV could replicate in the absence of sphingolipids, we used the small-molecule inhibitors myriocin and fumonisin B1 (FB1) to block the activity of SPT and CerS, respectively (Fig. 3a). Inhibition of sphingolipid synthesis was confirmed by pulse-chase experiments using the SM precursor [$^{14}$C]-serine (Supplementary Fig. 4a, b), and LC–ESI–MS/MS confirmed that inhibitor

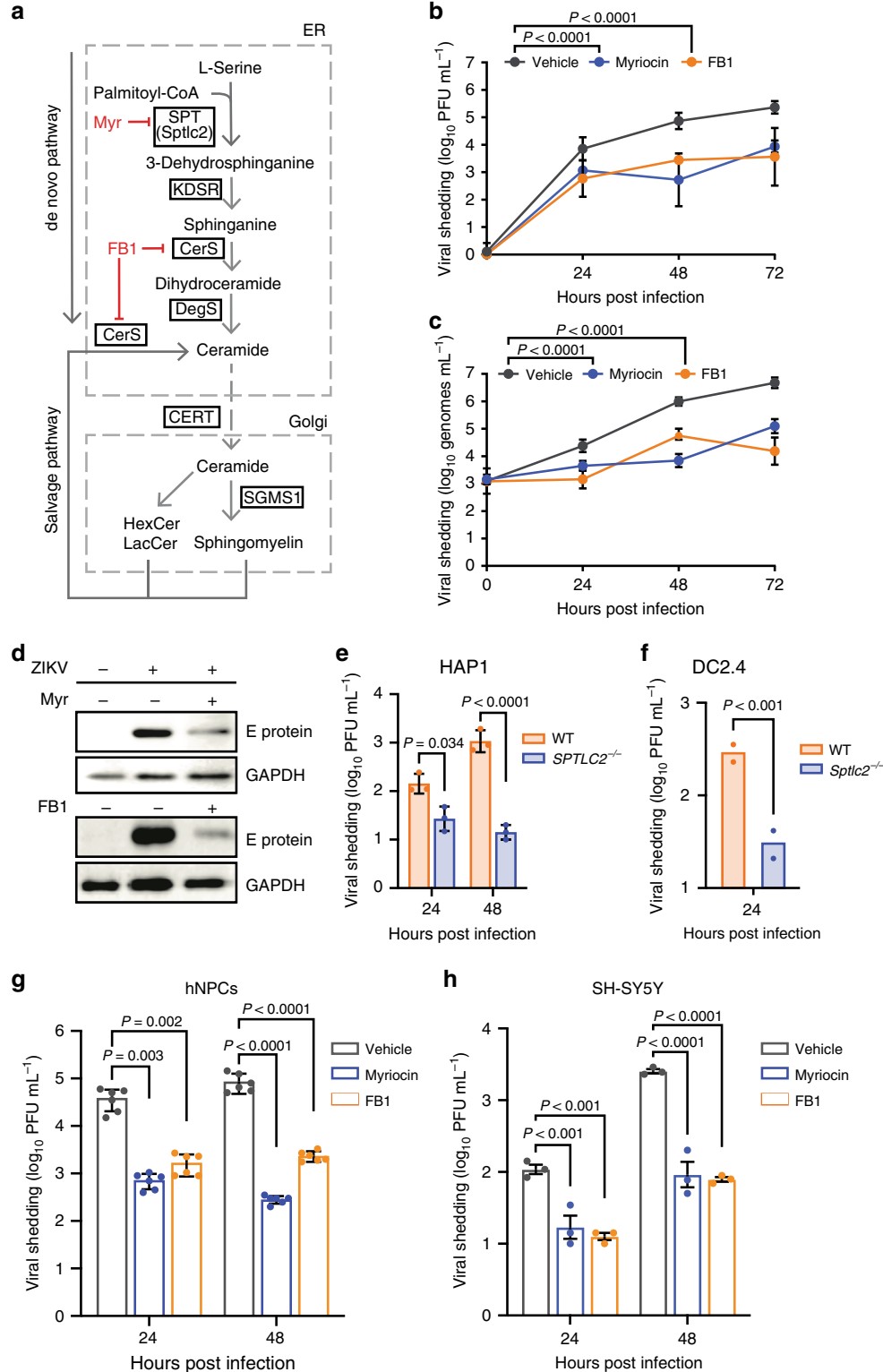

**Fig. 3 Sphingolipids are essential for ZIKV infection. a** Overview of sphingolipid biosynthesis. SPT serine palmitoyltransferase, KDSR 3-ketodihydrosphingosine reductase, DegS delta 4-desaturase, sphingolipid, CerS ceramide synthase, SGMS1 sphingomyelin synthase 1, CERT ceramide transfer protein. **b, c** Huh7 cells treated with myriocin, FB1, or a vehicle control were infected with ZIKV (MOI = 0.1). At the indicated times post infection, culture supernatants were collected and analyzed by plaque assay (**b**) or RT-qPCR (**c**); $n = 3$ independent experiments. Two-way ANOVA with Dunnett's multiple-comparison test. **d** Vero cells treated with myriocin, FB1, or vehicle were infected with ZIKV (MOI = 0.1). At 72 hpi, intracellular levels of ZIKV E protein were assessed by immunoblotting. Blot is representative of two independent experiments. **e, f** HAP1 human (**e**) and DC2.4 murine dendritic (**f**) WT and SPTLC2-knockout cells were infected with ZIKV (MOI = 0.1). At the indicated timepoints, culture supernatants were collected and analyzed by plaque assay; $n = 3$ independent experiments. Two-way ANOVA with Sidak's multiple-comparison test (**e**) and two-tailed Student's $t$ test (**f**). **g** iPSC-derived human neural progenitor cells (hNPCs) treated with myriocin, FB1, or vehicle were infected with ZIKV (MOI = 0.1). At the indicated times post infection, culture supernatants were collected and analyzed by plaque assay; $n = 6$ independent infections. Two-way ANOVA with Tukey's multiple- comparison test. **h** SH-5YSY human neuroblastoma cells were treated with inhibitors, infected with ZIKV (MOI = 0.1), and analyzed by plaque assay. Two-way ANOVA with Tukey's multiple-comparison test. Data are mean ± SD. See also Supplementary Data 3, Supplementary Fig. 4, Supplementary Fig. 5, and the Source Data file.

treatment effectively reduced overall levels of sphingolipids without perturbing levels of PS (Supplementary Fig. 4c–f), while cell growth rate and morphology were not affected (Supplementary Fig. 5a, b).

Next, we examined the ability of ZIKV to propagate in inhibitor-treated cells. Huh7 cells treated with myriocin or FB1 and infected at a MOI of 0.1 showed a 30-fold reduction in infectious virions released into the culture supernatant 24 h after infection, a difference that increased to over 100-fold after 72 h (Fig. 3b). To control for the possibility that the inhibitors reduced ZIKV particle infectivity, we performed real-time quantitative polymerase chain reaction (RT-qPCR) to quantify the number of ZIKV genomes released into the supernatant of treated and untreated Huh7 cells (Fig. 3c). Myriocin or FB1 treatment caused reductions in levels of extracellular viral RNA comparable to the losses in infectious titer we found by plaque assay, indicating that inhibition of sphingolipid biosynthesis caused a defect in ZIKV particle biogenesis. Similarly, propagation of ZIKV in myriocin or FB1-treated Vero cells, a primate epithelial cell line highly permissive to flavivirus infection[37], resulted in decreased load of E glycoprotein at 72 hpi (Fig. 3d).

To compare the effects of inhibiting sphingolipid metabolism to inhibiting other lipid metabolic pathways known to play a role in the flavivirus replication cycle[38], we treated Huh7 cells with lovastatin (inhibitor of HMGCR, a key enzyme in cholesterol biosynthesis) or PIK93 (inhibitor of phosphatidylinositol-4-kinase) before infection with ZIKV; in contrast to myriocin and FB1, lovastatin treatment caused a more modest 30% decrease in viral shedding by 24 hpi, and PIK93 did not appear to decrease infection (Supplementary Fig. 6a, b). We concluded that depletion of cellular sphingolipids blocked ZIKV propagation in human and nonhuman primate cells in a manner specific to that lipid class.

To confirm this finding in a genetic model, we tested ZIKV propagation in two cell lines negative for SPTLC2: human haploid HAP1 cells[39] with a CRISPR-mediated knockout ($SPTLC2^{-/-}$), and knockout murine dendritic DC2.4 cells ($Sptlc2^{-/-}$)[40]. HAP1 cells, which are generally excellent genetic models for studies of the virus life cycle[41–44], yielded a nearly hundred-fold decrease in viral shedding in SPTLC2$^{GT}$ cells by 48 hpi (Fig. 3e). Though immunocompetent murine cells are highly restrictive to ZIKV[45], we observed a tenfold reduction in ZIKV production from $SPTLC2^{-/-}$ DC2.4 cells at 24 hpi similar to that timepoint in our other cell lines (Fig. 3f); both WT and knockout DC2.4 cells appeared to clear the infection at later timepoints as previously reported in vivo[45], further suggesting that sphingolipids are required for fundamental aspects of flavivirus biology rather than interactions with specific cell types or antiviral immunity.

**Sphingolipids are required for ZIKV infection of neural progenitor cells.** ZIKV is known to be neurotropic and mainly infects neuronal cells within the central nervous system to cause microcephaly[2]. Recent studies have used iPSC-derived human neural progenitor cells (NPCs) for their studies as a proxy to physiologically relevant primary cells[3,46–48]. Therefore, we decided to use iPSC-derived NPCs to validate our key findings by manipulating sphingolipid levels in these cells using myriocin and FB1. Inhibitor-treated cells showed a significantly reduced level of viral shading as compared with untreated cells, showing roles for sphingolipids in ZIKV infection in neuronal primary cells (Fig. 3g). In addition to iPSC-derived NPCs, we also employed the human neuroblastoma cell line SH-SY5Y to validate our initial findings further. Consistent with the Huh7 and NPCs, both myriocin and FB1 significantly reduce the amount of ZIKV production (Fig. 3h) as compared with the control cells. These data demonstrate roles for sphingolipid in ZIKV infection and validate our initial observations.

**Sphingolipids are required for ZIKV replication.** Both the sphingolipid metabolic network and flavivirus replication cycle are distributed throughout the cell, leading us to investigate where and when the essential ZIKV–sphingolipid interactions suggested by our inhibitor experiments occur. To test whether sphingolipids are required for ZIKV binding or entry as has been reported for other positive-strand RNA viruses[49], we incubated monolayers of Vero cells pretreated with myriocin or FB1 with ~100 ZIKV PFUs on ice for 0–45 min, then washed away unbound virions, and overlaid the cells with methylcellulose overlay media that did not contain inhibitors. After 3 days, viable cells were stained with crystal violet, and numbers of plaques counted (Fig. 4a). Inhibitor-pretreated Vero cells showed no losses in absolute levels or the rate of plaque formation, indicating that myriocin inhibition of sphingolipid synthesis did not significantly affect ZIKV binding or internalization.

To investigate whether ZIKV requires sphingolipids during genome replication in the replication factory, we used RT-qPCR to compare levels of intracellular viral RNA in Huh7 cells treated with myriocin or FB1 relative to a vehicle control. Supporting the results of our entry assay, levels of ZIKV RNA bound to cells at 0 hpi were not significantly changed by sphingolipid depletion (Fig. 4b). After the initiation of RNA replication by 3–4 hpi, levels of RNA in sphingolipid-depleted cells more than 60% relative to vehicle-treated cells, remained over 30% lower by 20 hpi (Fig. 4b). To confirm that these differences were due to decreased RNA replication rather than defects at other stages of the viral life cycle, we treated infected sphingolipid-depleted and normal cells with an inhibitor of ZIKV RNA polymerase (TPB)[50], and measured intracellular replication relative to non-TPB-treated cells at 8 hpi

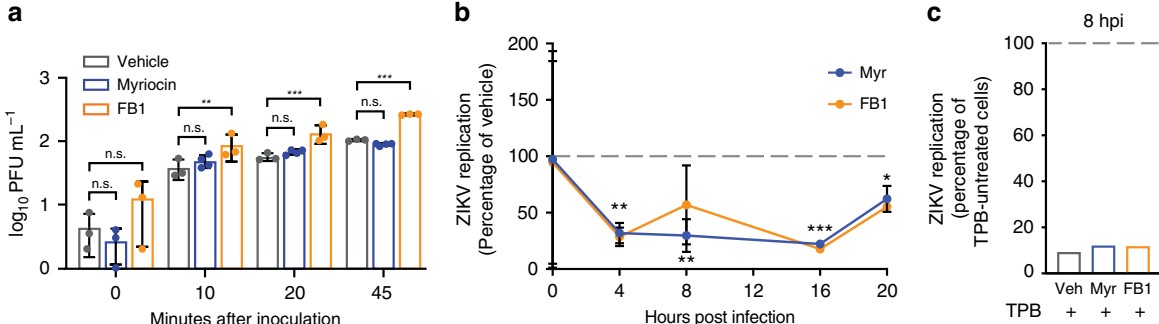

**Fig. 4 Sphingolipids are required for ZIKV replication. a** Monolayers of Vero cells pretreated for 3 days with myriocin, FB1, or a vehicle control were incubated on ice for the indicated times with 100 PFUs before washing and overlay with media without the inhibitors. Plaques were counted after 3 days. Data are representative of four independent experiments. Two-way ANOVA with Tukey's multiple-comparison test. **b** Huh7 cells pretreated for 3 days with myriocin, FB1, or a vehicle control were infected with ZIKV (MOI = 20). Intracellular ZIKV replication in myriocin or FB1-treated cells was measured at the timepoints shown and plotted relative to vehicle. $n = 3$ independent experiments. $n = 4$ independent experiments. **c** Huh7 cells treated with inhibitors as in (**b**) were pretreated for 24 h with the RNA polymerase inhibitor TPB, infected as before, then maintained in TPB and inhibitor/vehicle-treated media for 8 h. At 8 hpi, intracellular replication in TPB-treated cells was measured relative to non-TPB-treated cells for each condition. $n = 2$ independent experiments. Data are mean ± SD; n.s. not significant, $*P < 0.05$, $**P < 0.01$, $***P < 0.001$, two-tailed Student's $t$ test. See also the Source Data file.

(Fig. 4c). ZIKV RNA in both sphingolipid-depleted and vehicle cells decreased by approximately tenfold under TPB treatment, ruling out a prereplication viral requirement for sphingolipids. We concluded that sphingolipids are dispensable for ZIKV entry into host cells, but are required for viral replication.

**The Cer-SM network is a key determinant of ZIKV infection**. Of the thousands of known sphingolipid structures[51], which specific categories of species are required for ZIKV replication? Our findings that SM and Cer, despite opposing trends during infection, which are linked by a single enzymatic step in the sphingolipid metabolic network (Fig. 5a), prompted us to study their relationship more closely. First, we calculated ratios of normalized lipid levels for all possible pairs of SM and Cer species in our dataset (Supplementary Data 4), 382 of which varied significantly across experimental conditions ($P < 0.01$, one-way ANOVA). We then used recursive feature elimination with cross-validation (RFECV) to define a subset of 250 lipid pairs with the features of greatest biological interest (Supplementary Data 5). PCA of these ratios (Fig. 5b) revealed a pattern of infection- and time-dependent separation similar to the overall distribution of lipid species observed previously, indicating that the SM–Cer metabolic network is strongly perturbed by ZIKV infection. Furthermore, hierarchical clustering of the lipid pairs resulted in near-complete separation, as measured by identity of the nearest neighbor, by ratio type (Cer/Cer, SM/SM, or Cer/SM) (Fig. 5c). Remarkably, every Cer/SM ratio without exception decreased from 24 to 48 hpi in mock cells and increased in infected cells, suggesting that flux between ceramide and SM is highly regulated during ZIKV replication.

While the trend of ceramide enrichment relative to SM was consistent across ceramide species, there was considerable variation among ceramide species in both our ZIKV and NS4B experiments (Fig. 2d). To identify relationships of biological interest between ceramide species, we tested RFECV-identified ratios that fell above the stringent cutoffs of component score > 0.1 and $F$ score > 50. This yielded four Cer/Cer ratios ($P < 0.01$, one-way ANOVA) (Fig. 5d–g). Strikingly, each of these high-interest ratios contained the (dihydro)ceramide Cer(d18:0/16:0) or Cer(d18:1/16:0), which was the single most enriched lipid by both ZIKV and NS4B (Fig. 2d). Recently, it has been shown that the six mammalian ceramide synthases (CERS1–6) preferentially catalyze the formation of ceramide species with different acyl-chain lengths and degrees of saturation[52,53], and that the

signaling properties of ceramides are influenced by acyl-chain identity[54]. Our data indicate that ZIKV infection and NS4B expression increase the production of Cer(d18:1/16:0) relative to other ceramide species.

Three major pathways control ceramide levels in mammalian cells: (1) de novo synthesis in the ER, (2) degradation of SM by a family of sphingomyelinases (SMases), and (3) salvage of sphingolipid catabolism products via conversion to sphingosine (Fig. 5a). Because SM is the most abundant sphingolipid in mammalian cells and represents a major outlet for newly synthesized ceramides, we reasoned that flux between SM and Cer likely contributed to ZIKV-driven increases in Cer/SM ratios. To test this hypothesis, we infected KBM7 cells bearing a gene-trap mutation for SGMS1, the major human sphingomyelin synthase[55,56]. KBM7 SGMS1[GT] cells displayed higher Cer/SM ratios than did WT or reconstituted mutant cells (KBM7 SGMS1[GT] + SGMS1), recapitulating the trend seen in the Cer/SM ratios identified by RFECV[57]. Indeed, KBM7 SGMS1[GT] cells were over 100-fold more permissive to ZIKV infection than WT or SGMS1[GT] + SGMS1 cells (Fig. 6a), suggesting that even modest increases in pools of intracellular ceramide could dramatically enhance viral replication.

Having increased virus production by blocking the conversion of Cer to SM, we predicted that blocking the SMase pathway of Cer synthesis would have the opposite effect. When treated with the neutral SMase inhibitor GW4869, infected KBM7 SGMS1[GT] + SGMS1 cells displayed a 70% reduction in ZIKV shedding without altering cell viability, while SGMS1[GT] cells were unaffected (Fig. 6b, Supplementary Fig. 7a, b). To validate our KBM7 experiments in Huh7 cells, we again used GW4869 to block conversion of SM to Cer, and exogenously added recombinant SMase to increase Cer flux. GW4869 treatment reduced viral shedding, and SMase treatment modestly increased permissiveness to ZIKV infection (Fig. 6c, Supplementary Fig. 8a–d). Taken with our findings that depletion of all sphingolipids reduces viral replication (Fig. 6d), these data show that ZIKV targets the SM–Cer flux to successfully establish infection in the host cells.

Dysregulation of lipid metabolism contributes to diverse human diseases[58], including sphingolipidoses, such as Niemann–Pick disease and hereditary sensory neuropathy[59,60]. The physical interactions between lipids, proteins, and other macromolecules are increasingly the focus of efforts to systematically map the biological networks underlying disease[61,62].

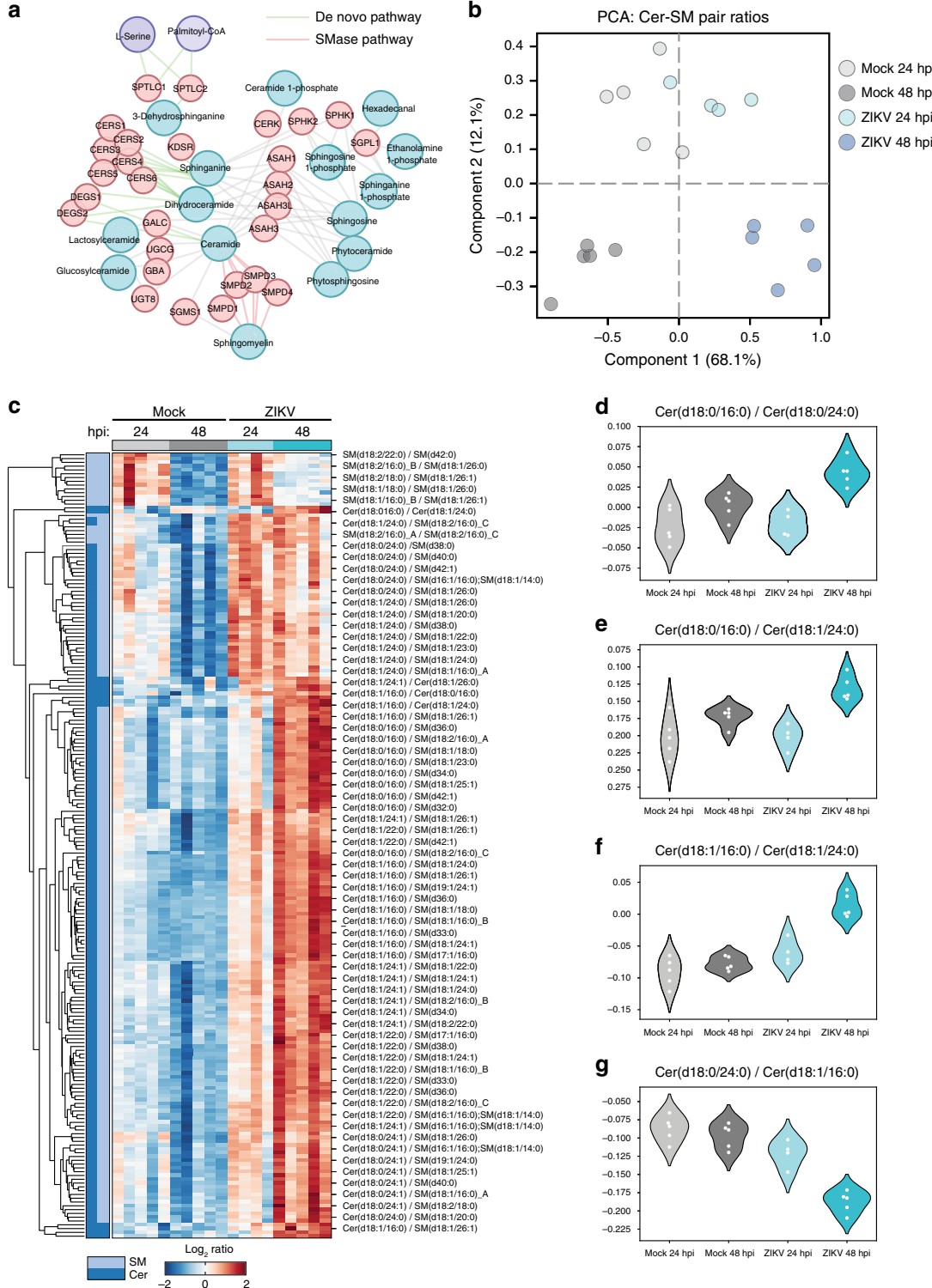

**Fig. 5 Targeted regulation of sphingolipid metabolism by ZIKV. a** Overview of the ceramide metabolism network. **b** Ratios of normalized lipid levels were calculated for all combinations of SM and Cer species in our dataset. Ratios that varied significantly across conditions (n = 382, P < 0.01, one-way ANOVA) were analyzed with recursive feature elimination with cross-validation, and the resulting high-interest Cer/SM pairs (n = 250) were analyzed with PCA. **c** Heatmap of clustered log2 SM/Cer ratios identified in (**b**). Each column represents a single biological sample. **d–g** Log2 ratios from (**b**) that met the following criteria: PC length > 0.1, F score > 50, and P < 0.01. SM sphingomyelin, Cer ceramide. See also Supplementary Data 4 and the Source Data file.

These maps have provided evidence that disorders, which are unrelated in origin, but similar in phenotype, can share overlapping patterns of genetic and metabolic perturbations, forming distinct disease modules within the human interactome[63–65].

Having shown the importance of the sphingolipid network in ZIKV replication, we asked if it interacted with known disease modules that were clinically similar to the symptoms of ZIKV infection. To accomplish this, we built a metabolic network containing the biosynthesis pathways of each subclass in our

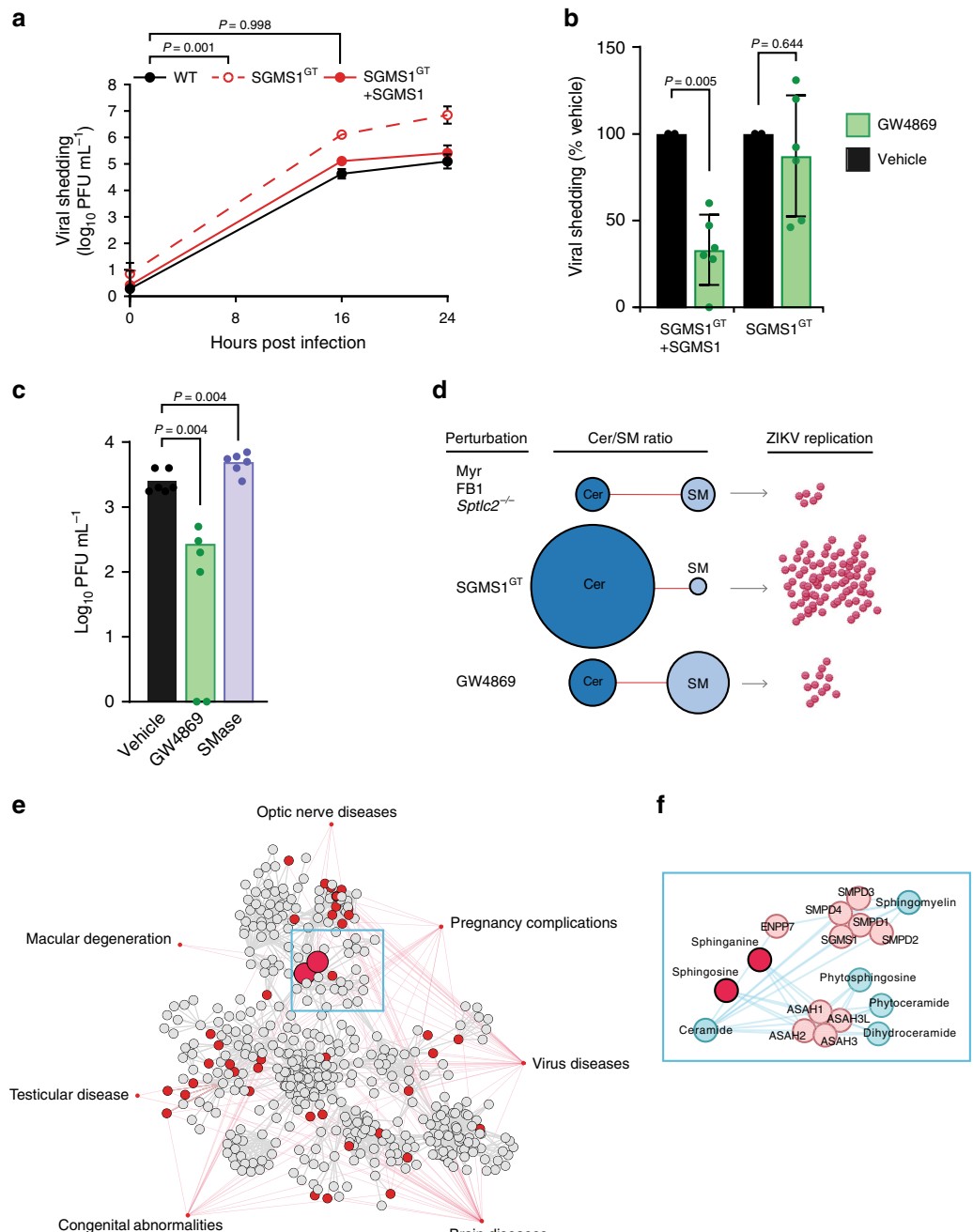

**Fig. 6 Elevated ceramide levels increase ZIKV infection. a** KBM7 WT, SGMS1[GT], and SGMS1[GT] + SGMS1 cells were infected with ZIKV (MOI = 1). At the indicated timepoints, culture supernatants were collected and titrated by plaque assay; $n = 8$ biological replicates. Two-way ANOVA with Dunnett's multiple-comparison test. **b** KBM7 SGMS1[GT] and SGMS1[GT] + SGMS1 cells were infected with ZIKV (MOI = 1) and treated with 10 μM GW4869 or vehicle. At 24 hpi, supernatants were collected and analyzed by plaque assay; $n = 6$ independent infections. Two-tailed Student's $t$ test. **c** Huh7 cells were infected with ZIKV (MOI = 1) and treated with 10 μM GW4869 or recombinant neutral sphingomyelinase (SMase). At 24 hpi, culture supernatants were analyzed by plaque assay; $n = 2$ independent experiments. One-way ANOVA with Dunnett's multiple-comparison test. **d** Model of experimental perturbations to the Cer/SM metabolic network and their effects on ZIKV replication. **e** Network of associations between disease modules similar to congenital ZIKV syndrome and lipid metabolism pathways. A metabolic network connecting the lipid subclasses identified by lipidomics was mapped to seven medical subject heading (MeSH) disease terms selected for their phenotypic similarity to clinical ZIKV syndrome. Nodes represent enzymes, lipids, and other metabolites in lipid biosynthesis, and gray lines represent reactions. Red nodes are metabolites associated with the MeSH ontologies linked by red lines. **f** Inset panel showing the metabolic neighborhood of the sphingolipids sphinganine and sphingosine. Data are mean ± SD. See also Supplementary Figs. 6–8, Supplementary Data 5, and the Source Data file.

lipidomics dataset (Supplementary Fig. 9), then surveyed the network for metabolites associated with seven Medical Subject Heading (MeSH) terms selected for their similarity to clinical outcomes of ZIKV infection[66–68]. As a consequence of including all compounds and genes from the constituent lipid biosynthesis pathways in our metabolic model (Supplementary Data 5), nearly every node of the resulting disease–metabolite network was a non-lipid compound capable of participating in other aspects of

cellular metabolism (Fig. 6e). Remarkably, however, the two network nodes with the greatest number of connections to MeSH disease terms were the sphingosine and sphinganine, which differ by only a single double bond (Fig. 6f). In addition to forming the sphingoid bases incorporated into all other sphingolipids, sphinganine and sphingosine are immediate neighbors of SM and Cer, with which they can be readily interconverted (Fig. 6f and Fig. 5a).

**Ceramide is redistributed to ZIKV replication membranes**. The set of network perturbations we defined left open the function of ceramide's role during ZIKV replication. Our results suggested that the defects in production of new ZIKV RNA and proteins in the replication factory (Figs. 3d, 4b, c), rather than their assembly or maturation (Figs. 3c, 4a), caused reduced viral shedding from sphingolipid-depleted cells, leading us to hypothesize that ceramide was physically recruited to membranes of the ZIKV replication site. In agreement with our lipidomics results, super-resolution microscopy revealed a close association between ceramide and RC marker NS4B (Fig. 7a, g).

To test whether this NS4B-ceramide recruitment was required for viral replication or a by-product of bulk membrane remodeling during infection, we transfected Huh7 cells with biosensors for PI (4,5)$P_2$ and PI4P, lipids in the upregulated PI signaling pathway (Figs. 1c, d, 2c), which had previously been indicated to participate in formation of positive-strand RNA virus replication membranes[69]. Indeed, PI4P—but not the PM-localized PI(4,5)$P_2$—colocalized with ZIKV NS4B by 24 hpi (Supplementary Fig. 10a, b). Having previously found that inhibition of PI phosphorylation had no effect on ZIKV replication (Supplementary Fig. 6a), we took these results to show that the recruitment and presence of ceramide in replication membranes are required for ZIKV replication in a manner outside of bulk recruitment. As a further control, we measured overlap between ceramide and E protein, which is produced at replication sites, but not thought to be present in replication vesicles; further supporting a specific function for ceramide in replication vesicles, we observed no correlation with E protein (Fig. 7b, g)[13].

Sphingomyelin is enriched in the *trans*-Golgi network, PM, and vesicle transport system[70], while the events of ZIKV replication occur in the ER. Because we had shown that hydrolysis of SM to ceramide by neutral SMase was partially responsible for the activity of ceramide during viral replication, we asked whether SM was recruited to replication sites to serve as a source of ceramide species during infection. To visualize the intracellular distribution of SM in mock and infected cells, we utilized Eqt-SM-GFP, a genetically encoded probe capable of binding diffuse pools of sphingomyelin[70,71]. We did not observe correlation between Eqt-SM-GFP and NS4B or E (Fig. 7c, d, g). This did not prevent enrichment of ceramide in the ER during ZIKV infection (Fig. 7e, f, h), suggesting that multiple pathways in sphingolipid biosynthesis and transport are coopted to support ZIKV replication.

## Discussion

Following the explosive outbreak of Zika virus in South and Central America and subsequent discovery of a startling array of novel virulence traits and modes of transmission, system biology approaches have provided a wealth of insights into the host–virus interactions underlying ZIKV disease. While these studies have opened promising avenues for further investigation, a major limitation is their focus on genetically encoded host factors, leaving the landscape of ZIKV–host metabolite interactions relatively uncharacterized. To fill in this gap in knowledge, we performed a global lipidomic survey in ZIKV-infected human cells, and found that ZIKV alters cellular sphingolipid metabolism to selectively increase production of ceramides, which are then enriched at viral replication sites. We confirmed that this relationship benefited ZIKV replication through targeted perturbations of sphingolipid metabolism in human neural progenitors and other cell types, resulting in enrichment or depletion of certain species, and corresponding increases or decreases in ZIKV RNA synthesis, virion biogenesis, and viral shedding. In summary, we have mapped a host–virus wiring diagram of sphingolipid metabolism in human cells.

Our findings present several avenues to fill in the circuitry of this wiring diagram and connect it to other maps of the molecular factors of ZIKV pathogenesis. From a mechanistic standpoint, it is now clear that individually expressed ZIKV proteins can regulate specific cellular processes[27,72–77], and that minor changes in genome sequence can greatly affect ZIKV infectivity and virulence[78–80]. We show here that ectopic expression of ZIKV NS4B, a transmembrane protein integral to the Flaviviridae RC[22], causes changes in sphingolipid metabolism similar to what we observed during full viral infection, implying a causal relationship between NS4B and dysregulation of sphingolipid homeostasis. It is notable that this relationship is specific to sphingolipids, as other regulations of other lipid classes showed only modest correlations between NS4B and ZIKV infection; future work could address whether similar links exist for other ZIKV NS proteins and lipid networks, and the extent to which these mechanisms are conserved across viral clades.

The biosynthesis, degradation, and intracellular distribution of ceramides is tightly controlled through a complex regulatory circuit[81–86], including glycosylation to form complex sphingolipids or conversion to the highly abundant sphingomyelin[87]. We observed increasing ratios of ceramide to sphingomyelin over the course of infection, essentially reversing homeostatic trends in both lipids. To test this experimentally, we infected a KBM7 cell line containing an inactivating mutation for SGMS1, the enzyme responsible for most SM synthesis in mammalian cells[55,88], and observed significant enhancement of ZIKV infection in SGMS1$^{GT}$ cells. Ceramide can be obtained through de novo synthesis in the ER or directly through the degradation of sphingomyelin by a family of SMases. We observed that GW4869-mediated inhibition of SMase activity decreased virus production in genetically complemented KBM7 SGMS1$^{GT}$ cells, while exogenous addition of SMase enhanced it; in contrast, uncomplemented KBM7 SGMS1$^{GT}$ cells were unaffected by either treatment. Therefore, the SMase degradation pathway represents a major pipeline in ZIKV ceramide recruitment. Further work is required to characterize how other modes of ceramide homeostasis, including conversion to complex sphingolipids, are coopted by ZIKV.

The challenges of studying lipidomic rather than genetic or proteomic factors of disease have been discussed in detail[89], yet the results we present here in human cells and by others in the flavivirus arthropod vector[90–92] argue strongly for the equal importance of lipids in questions of ZIKV pathogenesis. Our evidence that sphinganine and sphingosine are strongly linked to diseases with phenotypes that are clinically similar to ZIKV syndrome further implicates the sphingolipid network as contributing to the molecular processes underlying ZIKV virulence. Indeed, the role of sphingolipid signaling in regulating cell death, including during neural development, has been extensively described[93–95]. Further work is needed to validate lipidomic trends during ZIKV infection in clinical samples, as has recently been shown for Ebola virus[96,97]. A variety of nontoxic pharmacological modulators of sphingolipid metabolism exist, including one that is FDA-approved, raising the possibility that sphingolipid metabolism could be targeted as part of a host-directed therapeutic strategy in infected patients[98,99].

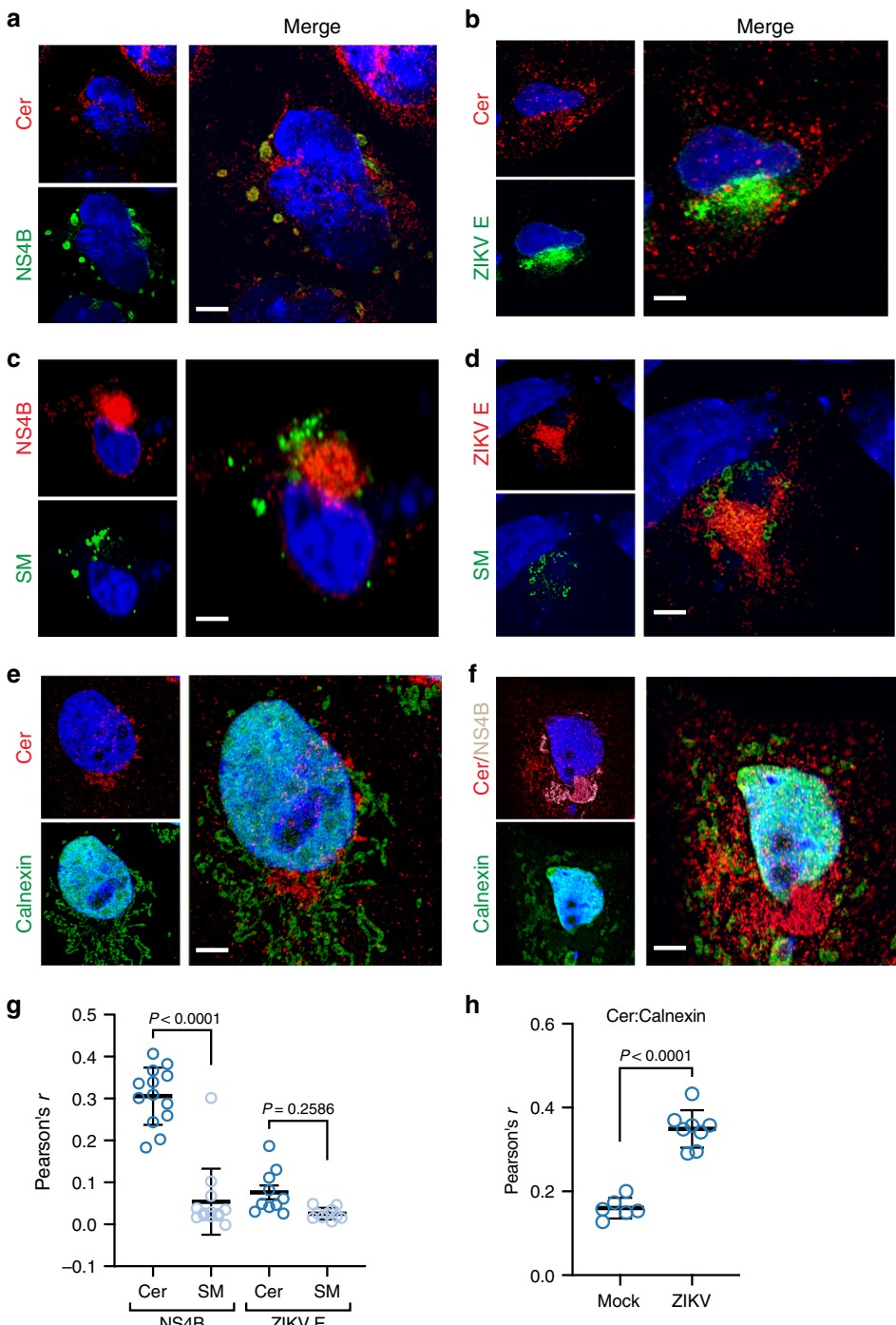

**Fig. 7 Ceramide redistributes to ZIKV replication sites.** Huh7 cells were infected with ZIKV (MOI = 10). At 24 hpi, mock and infected cells were fixed, co-stained with the indicated antibodies, and visualized with Airyscan superresolution light microscopy. All images are representative of four independent experiments. **a**, **b** Cells were co-stained with antibodies against ceramide and ZIKV replication marker NS4B (**a**) or E protein (**b**). **c**, **d** Huh7 cells were transfected with SM-Eqt-GFP immediately following infection, then fixed, stained with antibodies against ceramide and NS4B (**c**) or E protein (**d**), and visualized as before. **e**, **f** Mock (**e**) and infected (**f**) cells were stained with antibodies against ceramide, NS4B, and the ER marker calnexin. **g**, **h** Pearson's correlation coefficient was calculated for the indicated pairs of signals (n = 10 cells per condition pooled from four independent experiments; each dot represents r from a single field of view containing one or more cells). Thick central bar, mean; upper and lower bars, SD. Unpaired two-tailed Student's t test. **a–f** Scale bar, 10 μm.

## Methods

**Cell culture and transfections**. Huh7, Vero, and HEK 293T cells were maintained in DMEM; HAP1 cells and KBM7 cells were maintained in IMDM. Both media contained 10% FBS, 100 units/mL penicillin, 100 μg/mL streptomycin, and 1% nonessential amino acids. C6/36 cells were maintained in MEM containing 5% FBS, 100 units/mL penicillin, 100 μg/mL streptomycin, and 1% nonessential amino acids. Mammalian cell lines were maintained at 37 °C and 5% $CO_2$; C6/36 cells were maintained at 32 °C and 5% $CO_2$.

**Transfections**. C-FLAG pcDNA3 plasmids containing ZIKV NS4B were generously provided by the Alec Hirsch. Low-passage HEK 293T cells were seeded at a

density of 350,000 cells per well in 6-well plates. Transfections were carried out the next day using Lipofectamine 3000 transfection reagent (Invitrogen) as per the manufacturer's suggested protocol, with 5 μg of plasmid DNA used per transfected well. Wells were transfected with C-flag NS4B vector or with equal quantities of empty vector. Media was changed 6 h after transfection. After 48 h post transfection, individual wells were harvested for lipid purification as in the lipidomics sample preparation protocol. Cells were also fixed with 4% PFA, then stained with an Alexa Fluor 488-conjugated anti-FLAG antibody (Cell Signaling Technology #5407) to confirm high levels of transfection efficacy. For experiments with the Eqt-SM-GFP sphingomyelin probe, GFP-P4M-SidM PI4P probe, and pLNCX-PH-PLCδ1-GFP PI(4,5)P$_2$ probe, Huh7 cells were seeded on glass coverslips in a 24-well plate at a density of 25,000 cells per well. Transfections were carried out with Lipofectamine 3000 transfection reagent according to the manufacturer's instructions; at 24 h post transfection, cells were mock-infected or infected with ZIKV.

**Stem cells**. Human iPSC (XCL-1)[100]-derived neural progenitor cells (hNPCs) were purchased commercially (StemCell Technologies, Cat #70901, Lot #17080) and maintained up to ten passages according to the manufacturer's instructions. Briefly, hNPCs were grown on Matrigel-coated culture plates in Neural Progenitor Medium 2 (StemCell Technologies) up to 95–100% confluency, then detached using Accutase, and seeded at a density of 80,000–100,000 cells/cm$^2$.

**Viruses**. Zika virus strain FSS1302536 was propagated in C6/36 or Vero cells, and viral stocks and experimental samples were titrated on Vero cells by plaque assay (ZIKV) as described previously[101].

**Infections**. For infections of Huh7, Vero, or DC2.4 cells, ZIKV stocks were diluted to the desired MOI in DMEM or RPMI containing 2% FBS, and added to cells for 1 h at 37 °C and 5% CO$_2$ with constant rocking. The virus-containing supernatants were then aspirated, and the infected cells washed three times with PBS before addition of culture media and continued growth under normal conditions until the desired timepoints. For KBM7 suspension cells, 500,000 cells per replicate were centrifuged for 5 min at 300×g at room temperature, and resuspended in 1 mL of IMDM containing 2% FBS and ZIKV. Cells were rocked for 1 h at 37 °C and 5% CO$_2$ in 6-well plates, then centrifuged as before, and resuspended in PBS three times. After aspiration of the final wash, cells were resuspended in culture media and grown under normal conditions until the desired timepoints.

**Lipid sample collection**. Huh7 cells were seeded at a density of 1.2 million cells in 15-cm dishes and infected the next day with ZIKV FSS13025 at an MOI ~50 PFU/cell to ensure that all cells were infected. At 24 and 48 hpi, mock and infected cells were washed three times with ice-cold PBS and detached by scraping. A small fraction of the cell suspension was retained for protein content determination by BCA assay (Supplementary Data 1); the remaining volume was transferred into glass sample tubes on ice and centrifuged for 5 min at 500×g and 4 °C. After aspiration of the PBS supernatant, cell pellets were resuspended with 1 mL of ice-cold methanol and stored at −80 °C. Using a modified Folch extraction[102], chloroform and water were added to samples for a final ratio of 8:4:3 chloroform:methanol:water. The samples were vortexed to mix, chilled on ice for 5 min, and then vortexed again. The samples were incubated at 4 °C for 2 h to allow for the separation of the phases. The lower organic lipid-containing layer was removed, dried in vacuo, and then stored at −20 °C in 2:1 chloroform:methanol (v/v) until analysis.

**LC-MS/MS analysis and lipid identification**. LC–MS/MS parameters and identifications were conducted as outlined[103]. A Waters Aquity UPLC H class system interfaced with a Velos-ETD Orbitrap mass spectrometer was used for LC–ESI–MS/MS analyses. Lipid extracts were dried in vacuo, reconstituted in 10 μl of chloroform plus 540 μl of methanol, and injected onto a reverse-phase Waters CSH column (3.0-mm × 150-mm × 1.7-μm particle size), and lipids were separated over a 34-min gradient (mobile phase A: ACN/H$_2$O (40:60) containing 10 mM ammonium acetate; mobile phase B: ACN/IPA (10:90) containing 10 mM ammonium acetate) at a flow rate of 250 μl/min. Samples were analyzed in both positive and negative ionization modes using higher-energy collision dissociation and collision-induced dissociation to obtain high coverage of the lipidome. The fragment ions used for lipid identifications were used as previously outlined[103]. The LC–MS/MS raw data files were analyzed using LIQUID whereupon all identifications were manually validated by examining the fragmentation spectra for diagnostic and fragment ions corresponding to the acyl chains. In addition, the precursor mass isotopic profile and mass ppm error, extracted ion chromatograph, and retention time for each identification was examined. To facilitate quantification of lipids, a reference database for lipids identified from the MS/MS data was created, and features from each analysis were then aligned to the reference database based on their identification, m/z, and retention time using MZmine 2[104]. Aligned features were manually verified, and peak apex-intensity values were exported for statistical analysis.

**QC, normalization, and statistical comparison methods**. Data from positive and negative ion modes were analyzed separately using MATLAB version R2016b (MathWorks). Any unobserved lipid values were recorded as missing (NAs), and the data were log$_2$-transformed. The rMd-PAV algorithm[105] was used to identify potential outliers on the basis of their correlation, median absolute deviation, and skew; confirmation of outlier biological samples was achieved via Pearson correlation between the samples. All lipids were assessed for having at least two observations across all samples and enough observations for performing either qualitative or quantitative statistical tests[105]; none of them failed to meet these requirements, and thus all lipids were retained for further analysis. The data were normalized using global median centering, in which each sample was scaled by the median of its observed abundance values. This approach has been described previously[106–109] and utilized in a number of other studies[110–112]. Lipids were evaluated using ANOVA with a Dunnett test correction to compare infected with mock at each timepoint (24 h and 48 hpi). Yellowbrick[113] was used to perform recursive feature extraction.

**Cell culture treatments**. Myriocin, FB1, GW4869, lovastatin, and PIK93 were purchased from Cayman Chemical. Myriocin was dissolved in DMSO to make 15 mM stock solutions, and added to culture media at 1:500 for a final concentration of 30 μM; because myriocin is not fully soluble at 15 mM, stocks were thawed at room temperature, then heated for 15 min at 55 °C immediately before addition to media. FB1 was dissolved in 1:1 acetonitrile:DI water to make 5 mM stocks, then added 1:1000 to culture media for a final concentration of 5 μM. Myriocin and FB1 treatments were carried out for 72 h before experimental manipulations; we did not observe significant differences in growth rate or morphology over that period (Supplementary Fig. 2). GW4869 was added to culture media for 24 h at a final concentration of 10 μM, and stocks were prepared as previously outlined[114]. No significant toxicity was observed by CellTiter Glo assay (Promega) (Supplementary Fig. 4). Lovastatin and PIK93 were dissolved in DMSO, tested negative for cytotoxicity by CellTiter Glo assay, and used to pretreat cells for 24 h before infection at the indicated concentrations (Supplementary Fig. 6). All inhibitor stocks were stored at −20 °C. Neutral sphingomyelinase from *Bacillus cereus* (Sigma-Aldrich) was supplied at 2.5 units/mL (one unit is defined as hydrolyzing 1 μM of TNPAL-sphingomyelin per min at pH 7.4 and 37 °C) in 50% glycerol containing 50 mM Tris-HCl, pH 7.5. SMase was diluted in culture media to a final concentration of 0.1 units/mL and added to cells immediately after infection with ZIKV before supernatants were collected for titration by plaque assay at 24 hpi.

**Sphingomyelinase activity assay**. Huh7 cells were seeded in 96-well plates at a density of 20,000 cells per well. Cells were treated with GW4869 as described above, then lysed in PBS with 1% Triton X-100 and 1 mM PMSF. Sphingomyelinase activity in cell lysates in the presence and absence of 10 μM GW4869 was determined with the Amplex Red Sphingomyelinase Assay (Invitrogen) as per the manufacturer's instructions.

**RT-qPCR**. To measure intracellular viral replication, Huh7 cells were treated with 30 μM myriocin, 5 μM FB1, or 1:500 (v/v) DMSO for at least 72 h before seeding in six-well plates (200,000 cells per well). Cells were incubated for 24 h in freshly treated media containing 20 μM TPB47 (ChemBridge) or an additional DMSO vehicle control, then infected with ZIKV at a MOI of 20. At 0, 4, 8, 12, and 24 hpi, cells were lysed with Trizol reagent (Invitrogen) and stored at −80 °C. RNA was harvested using a Trizol phenol–chloroform extraction according to the manufacturer's protocol, and converted into cDNA with a High-Capacity RNA-to-cDNA kit (Applied Biosystems) using random hexamers. Real-time PCR was performed with a StepOnePlus Real-Time PCR system (Applied Biosystems) using TaqMan primer/probe sets (Thermo Fisher) against ZIKV48 (Assay ID APH6AE9) and beta-Actin (Assay ID Hs99999903_m1) according to the manufacturer's protocol. ZIKV signal was normalized to beta-Actin, and relative comparisons between treatments were made with the 2$^{-\Delta\Delta CT}$ method[115]. For measurements of viral shedding, RNA was isolated from culture supernatants with Trizol according to the manufacturer's protocol, and absolute quantification was performed by constructing a gBlock (IDT) standard curve as described previously[116].

**Fluorescence microscopy**. Huh7 cells were seeded on glass coverslips in 24-well plates at a density of 20,000 cells per well and grown overnight. The next day, cells were infected with ZIKV at a MOI of 10 and/or transfected as needed, and grown another 24 h before fixation with 4% paraformaldehyde in PBS for 15 min at room temperature. After washing with PBS, cells were blocked/permeabilized in blocking buffer (10% normal goat serum, 0.1% Triton X-100 in PBS) for 1 h. Incubations with primary and Alexa Fluor-conjugated secondary antibodies diluted in blocking buffer (see "Reporting summary" for a list of antibodies and dilution information) were performed for 1 h at room temperature, separated by three 5-min washes with PBS. DAPI (4′,6-diamidino-2-phenylindole) was used to visualize nuclei. Coverslips were mounted on glass slides with ProLong Glass antifade reagent (Thermo Fisher). Images were acquired with a Zeiss LSM 880 laser-scanning confocal microscope in Airyscan mode using a 63×/1.4 NA oil objective. Fluorophores were excited sequentially by 405-/488-/561-/633-nm lines, and imaging conditions were

optimized to minimize bleed-through. Z stacks were taken with a 0.18-µm interval between slices. Airyscan processing was performed in Zeiss Zen software using the default settings.

**Image analysis**. Pearson's correlation coefficients (PCC) were measured from z stacks in Imaris 9.3.1 (Bitplane) using the included Coloc module, after thresholding the signals to reduce background as indicated in the Source Data. PCC values were calculated from at least ten cells per condition, pooled from two or three independent experiments.

**Immunoblotting**. Vero cells were seeded in 10-cm dishes at a density of 1 million cells per dish and infected the next day with ZIKV at a MOI of 0.1. At 48 and 72 hpi, cells were washed three times with PBS to remove extracellular virus and lysed with 1% SDS in PBS, followed by three rounds of heating (95 °C for 5 min) and manual vortexing to complete lysis. Protein levels were determined on a NanoDrop spectrophotometer (Thermo Fisher Scientific); equal amounts of protein were subjected to sodium dodecyl sulfate polyacrylamide gel electrophoresis, and transferred to a PVDF membrane (Amersham). Samples were blocked with 5% dried milk in TBST and probed with monoclonal antibodies for flavivirus E protein (4G2) (1:500) and GAPDH (1:1,000), followed by incubation for 1 h with HRP-conjugated secondary antibodies (1:1,000). Blots were visualized with SuperSignal West Pico chemiluminescent substrate (Thermo Fisher Scientific) on an ImageQuant LAS 4000 imager (GE Life Sciences).

**Binding and entry assay**. Vero cells were seeded in six-well plates at approximately 250,000 cells per well, so they were 80–90% confluent the next day. These monolayers were washed three times with ice-cold PBS to inhibit endocytosis, and inoculated with 100 PFUs of ZIKV diluted in ice-cold Opti-MEM containing 2% FBS (0.5 mL of inoculum per well). Cells and virus were incubated at 4 °C with rocking. After 0, 10, 20, and 45 min, the inoculums were aspirated, and the cell layers washed three times with ice-cold PBS before addition of Opti-MEM containing 2% FBS and 1% methylcellulose. Plates were incubated for 3 days at 37 °C and 5% $CO_2$. After that period, the overlay media was removed through repeated PBS washes, and the cells were fixed with 4% paraformaldehyde and stained as for plaque assays as described above.

**Metabolite–disease module network**. To generate a Cytoscape network model of the lipid subclasses identified in our lipidomics experiment, we input a curated list of gene or compound identifiers from KEGG (Supplementary Data 5) into the MetScape 3 Cytoscape app as previously described[117,118]. To determine whether any of the network nodes had been associated in the literature with disorders that were phenotypically similar to ZIKV syndrome, we used the MetDisease Cytoscape app[119] to query the Metab2MeSH database[120] for seven Medical Subject Heading (MeSH) terms ('macular degeneration', 'optic nerve diseases', 'pregnancy complications', 'virus diseases', 'brain diseases', 'congenital abnormalities', and 'testicular disease') arbitrarily selected for this purpose.

**Flow cytometry**. Huh7 cells were seeded at a density of 350,000 cells per well in six-well plates, then incubated in untreated DMEM or DMEM containing GW4869 at a concentration of 10 µM as above. After 18 h of treatment, Huh7 cells were harvested for staining and flow cytometry analysis. Cells were washed with PBS, stained with Zombie Violet Fixable Viability Kit (BioLegend) for 10 min. For ceramide labeling, cells were fixed with 4% PFA at RT for 15 min, blocked and permeabilized for 10 min with blocking buffer (FCS 5%, Triton X-100 0.1%, and PBS), and then stained for 30 min at RT with mouse anti-ceramide primary (1:100 in blocking buffer) and Alexa Fluor 555 anti-mouse secondary (1:100 in blocking buffer). Alexa Fluor 555 secondary staining was used as an isotype control. All samples were washed twice with FACS buffer before analysis. Data acquisition was carried out with BD FACSymphony flow cytometer with CellQuest (BD Bioscience) software; FlowJo (FlowJo LLC) software was used for data analysis and figure construction.

**Statistics**. Analysis of lipidomics datasets was performed as described above. All other statistical analyses were performed using Prism 8.0 (GraphPad). Unless otherwise stated, P values are from unpaired two-tailed statistical tests without adjustments for multiple comparisons, with $P < 0.05$ considered statistically significant.

**Reporting summary**. Further information on research design is available in the Nature Research Reporting Summary linked to this article.

## Data availability

Mass spectrometry datasets have been deposited at the Mass Spectrometry Interactive Virtual Environment (MassIVE) at the University of California, San Diego under the accession code MSV000085584 [https://doi.org/10.25345/C5HT4Z]. Source data are provided with this paper.

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

## Acknowledgements

This work was supported by the Collins Medical Trust and NIH grant R21AI133631 (H. C.L., J.B.W., and F.G.T.); NIH grant U19AI106772 and administration supplement 5U19A1106772-05 (J.E.K., K.G.S., J.-Y.L., and T.O.M.); NIH grant R21 AI135537, National Center for Advancing Translational Science CTSA UL1 TR000128, Oregon Clinical and Translational Research Institute (W.B.M.); NIH grants R01GM060170 and 9R01AI152579 (E.B.); NIH grants R01DK105961, U19AI106754, and U54GM069338 (A. R.N. and E.A.D.). Lipidomics analyses were performed at the Pacific Northwest National Laboratory in the Environmental Molecular Sciences Laboratory, a national scientific user facility sponsored by the U.S. Department of Energy (DOE) Office of Biological and Environmental Research. Reagents were generously shared by Alec J. Hirsch and Daniel N. Streblow (OHSU Vaccine and Gene Therapy Institute), and by Chris G. Burd (Yale), who provided Eqt-SM-oxGFP. We wish to thank Shandee D. Dixon, Fabian Pott, S. Farley, Ayna Alfadhli, Bettie Kareko, and Zoe C. Lyski (OHSU), Christopher J. Parkins and Jessica L. Smith (OHSU Vaccine and Gene Therapy Institute), and Erika M. Zink (PNNL) for providing technical support; Stefanie K. Petrie, Aurelie Snyder, and Brian Jenkins (OHSU Advanced Light Microscopy Core) for their help in designing and analyzing the confocal microscopy experiments; Lenette L. Lu (University of Texas Southwestern Medical Center) for substantial comments on the paper.

## Author contributions

F.G.T. conceived the study; F.G.T. and H.C.L. designed the experiments; J.E.K., K.G.S., L. M.B., and J.-Y.L. performed and analyzed the lipidomics experiments with input from T. O.M.; J.B.W. performed the flow cytometry and transfections; J.B.W. and H.C.L. performed the RT-qPCR; D.K. and J.B.W. performed the immunoblotting and microscopy experiments; A.R.N. and E.A.D. performed the pilot lipidomics experiments, which were analyzed by E.G.T.; T.H. provided the HAP1 mutant cell lines; J.B.W. and H.C.L. performed infection assays, cell culture, and propagated viruses; H.C.L. performed Airyscan microscopy and prepared the Cytoscape model; H.C.L. prepared figures and wrote the paper with input from W.B.M., E.B., and F.G.T.

## Competing interests

The authors declare no competing interests.
