## [Peer Review File · Nature Communications]

Reviewers' Comments:

Reviewer #1:

Remarks to the Author:

In this study, Leier et al examine the changes in the lipidome during infection with Zika virus (ZIKV). They find that the largest changes are in sphingolipids, and then they provide data that increased ceramide sensitizes cells to ZIKV infection and mediates ZIKV-induced apoptosis. The lipidomics results appear robust and quite informative. The approaches to implicate ceramide/sphingomyelin also provide strong data but could be improved.

Major Points

1. What's primarily missing from this study is any insight on how ZIKV modulates sphingolipids. Does the virus effects a change in metabolism AND in subcellular localization? Are the effects on ceramide directly related to the localization effects on SM or are these two independent effects? Does ZIKV bind to SM, does it change the localization of SMS? Does it in addition inhibit ceramide incorporation into complex sphingolipids? These points should be at least discussed.

2. The results in Figure 6 are very interesting results. Figure 6b plots cer/SM ratios for all species and the authors appear to suggest that this ratio is important in the regulation of ZIKV replication. It seems to this reviewer that the levels of ceramide are the ones that track best with infectious ability (more infection with high ceramide and less with less). This conclusion (if the authors agree), should be more clearly articulated. Also, did the authors measure sphingolipids, (CER and SM) after GW treatment and in SGMS1gT cells? Some kind of confirmation would be good. This is a key point that needs to be defined.

3. The colocalization studies in Figure 3 are interesting, but they use a discordant set of probes, and the authors may be misconcluding about ceramide. Eq-SM-GFP will stain endogenous SM. C6-NBD ceramide does not stain endogenous ceramide, and it is reported and used to stain Golgi. Importantly, C6-NBD ceramide may not reflect ceramide as this can be metabolized to SM. As such, it may be staining compartments of SM formation. This should be clarified using appropriate manipulations (e.g. inhibition of SM synthesis and effects on localization of the NBD-Cer). This will affect major conclusions in Figures 3 and 5. This also affects the rationale presented on top of page 20. It actually may be even more interesting if NBC-Cer is tracking endogenous SM. That will allow improved investigation of the effects of ZIKV on SM and SMS.

The authors should also note that Eq-SM-GFP may not have access to all endogenous SM, depending on exposure of SM and localization of the probe. Later the authors show many treatments that reduce SM; do these treatments reduce Eq-SM staining?

PIP2 localizes differently in E vs Gag expressing cells. Is Gag driving this change in localization?

Images are poor quality and often overexposed, which can mislead colocalization analysis.

Importantly, the quality of the panels in Figure 3 is not optimal. The authors should also show larger fields of cells.

Specific Points

4. The studies from Figure 1 clearly show major changes in sphingolipids that warrant further investigation based on those results; however, the presentation of Figure 2 and significance of the results are not clear to this reviewer. The authors seem to be making a point that sphingolipids are more worthy of investigation compared to other lipids based on the results in figure 2c? but this figure is not well explained, and the rationale of what makes the broad correlations of sphingolipids more important is not clear.

5. It is not clear what the parameters were that governed the choice of diseases similar to Zika (figure 4). if the study seeks changes caused by the virus, it is not clear why these changes should be similar in other diseases only because the phenotype is 'similar'. Also, if the network analysis in figure 4 is

being used as a probing experiment or as further 'fishing' for interactions of ZIKV with sphingolipids, then in that case it should be presented before Figure 3.

Why did the authors treat for 4 days with FB1 and MYR? It would appear that shorter time points may suffice. Were the SL levels studied in a time course experiment to determine the best time for treatments?

Also, did the authors culture these cells in serum (that would contain SM)? Perhaps this accounts for some of the observed effects on SM levels.

Importantly, can exogenously applied sphingolipids rescue the effects of FB1/MYR on virus shedding? The authors appear to indicate that exogenous SM and Cer (Figure 5) can reach the virus? So do they make up for the effects of Myr/FB1?

6. Figure 5. The CER/SM-Bodipy correlation with ER markers are not convincing. Anyway, one has no expectation for SM to localize to the ER.

7. While the analysis shown in Figure 7 is very nice, its conclusions are rather speculative, and is not a 'good note' to end on. Perhaps these results could be absorbed by figure 1 where changes in lipids are reported or at least moved up front?

Minor Points

1. Figure 4B. 'trivial' name is 3-ketosphinganine or 3-ketodihydrosphingosine, not 3

Dehydrosphinganine

2. Not clear why the authors used 30uM myriocin. This compound is potent and specific, and it inhibits SPT at 100-500nM.

3. Line 331-332; not clear; is there a decrease of 60%?

4. The use of plotting tools is impressive, however some times excessive or not fully informative. For example, the radar plots in figure 1 don't appear to be very informative. The bubble plots don't inform which individual species are in each bubble. Perhaps some labeling of some of the more outstanding ones would be helpful. Alternatively, perhaps the authors can use a heatmap.

5. Fig. 4d why is there virus shedding at time 0 after infection? Is this what is shown?

Reviewer #2:

Remarks to the Author:

Thank you for asking me to review the work entitled "A global lipid map reveals a network essential for Zika virus replication" by Leier et al. In this study, the authors investigated the global lipid profile of ZIKV-infected cells. They demonstrated that ZIKV remodeled the host lipid profile after infection and that virus replication can be inhibited by modulating the ratio of the intracellular lipid species. The authors further sought to establish a link between the identified ceramide with ZIKV-induced apoptosis, which has been suggested as a contributing factor to ZIKV-induced brain/testis damage. The team has a strong record in lipid research. However, the virology of the paper is not sufficiently supported by experiments and the overall data suffers from a lack of novelty. Moreover, a significant amount of essential (raw) data were not provided and a number of details concerning the lipidomic analysis require clarifications.

Major concerns:

1. The novelty and significance of the study is limited. Lipid is already known to be required for flavivirus replication and ceramide is known as a pro-apoptotic factor. This study did not provide enough significantly new information in this aspect.

2. The analysis is not well supported by validation experiments and the presented data lacks physiological relevance. Most of the validation experiments are done in Huh7 or VeroE6 cells, which are not the most biologically relevant cells for ZIKV infection in vivo; and it is particularly problematic with Huh7 (human hepatoma) cells which has a largely changed metabolic system as a cancer cell

line. Physiologically more relevant primary cell line and/or ex vivo organ culture models, and especially animal models must be applied to demonstrate the role of ceramide. More gene depletion experiments should be done to complement results from using inhibitors.

3. Line 119: what were the rationale and the viral replication kinetics for using 24 and 48 hpi? What were the cell conditions at the indicated time-points? Did the authors have a chance to compare lipidomic landscape under high and low MOIs?

4. Figure 2b and Figure 7c: the titles are missing in the X-axis.

5. Figure 3. It is not appropriate to use HIV Gag as a control for ZIKV prM+E. HIV Gag+Env should be used in this case. If immunostaining is against ZIKV E, the counter staining should be on HIV Env.

6. Figure 4f: explain why in Sptlc2^{-/-} cells, the ZIKV virus titer in 24hpi was even lower than that of the baseline at 0 hpi? It would be important to show the data at later time-points like 48 and 72hpi.

7. Figure 4b: ZIKV rescue assays must be performed to determine if addition of supplementary ceramide or other products along the pathway can reverse the anti-ZIKV effect of Myr and/or FB1.

8. Sample collection:

- Why were large dishes (15cm) used to culture low intensity cell (1.2×10^6 cells) – what are the advantages of this procedure?

- More detailed description about the protein normalization procedures must be provided. The normalization is an essential step with high MOI infection (with an MOI > 50 PFU/cell) and the infected cells should be dead at 24 and 48hrs. So when the cells were collected, the cell number must be very different between mock and infected groups, which will strongly influence the subsequent selection of significantly perturbed lipids.

9. Lipid extraction:

- The lower organic lipid containing layer was removed, dried in vacuum. Why was it stored at -20°C in 2:1 chloroform:methanol (v/v) until analysis (line 587)? Lipids extract was dried in vacuo and reconstituted in 10 µl chloroform plus 540 µl methanol (line 591). Why was it necessary to perform storage in 2:1 chloroform:methanol (v/v) if the final reconstituted solvent was 10 µl chloroform plus 540 µl methanol?

10. LC-MS/MS analysis and lipid identification:

- Were the corresponding internal standards included during the sample preparation and LC-MS analysis? These could not be found in the main text.

- The authors must provide the table of the 340 identified lipids.

10. Supplementary data1. Total protein content of cell lysate used for lipidomics: the protein concentration data in different cell samples could not be found. The authors must show these data and demonstrate how was the protein concentration used to normalize the MS intensity of individual sample (if total intensity normalization was used)?

11. Supplementary data2. The authors mentioned that 340 lipids were identified and these lipids' changes in normalized abundance between mock and infected cells could be calculated (line 118 to 120 in main text). However, these corresponding data were not found in the supplemental data 2 - only the column header and descriptions were shown.

12. Supplementary data3. The significant lipids and corresponding information such as ionization mode, associated adduct, m/z, retention time (RT), p-values, and log 2 fold change (FC) could not be observed in the supplemental data 3.

13. Supplementary data 5. The raw data of calculated ratios of normalized lipid levels for all possible pairs of SM and Cer species also could not be found in the Supplementary Data 5. The raw lipids table containing the calculated ratios values and statistical values must be shown.

Reviewer #3:

Remarks to the Author:

Leier et al analyzed the changes in lipid composition after injection with ZIKV. They identified

sphingolipids as an important mediator of virulence and provided information on the role of sphingolipids in viral replication.

My enthusiasm on the state-of-the-art lipid measurements are hampered by important details needing clarification and the use of fluorescently labeled lipid to study lipid localization. Please find my questions and recommendations on these below:

1. Lipid localization: Previous studies have already linked sphingolipids to ZIKV infection, as such, the novelty of the work mostly lies on the lipid colocalization with viral replication. Although fluorescent lipid analogs can be useful in studying more dynamic questions, the colocalization experiment can be (and in my opinion should be) carried out using probes with less perturbation on lipid structure such as clickable lipids. These lipids (alkynyl versions) are commercially available, can be used in fixed and live cells and exhibit a strong structural similarity to the endogenous lipids, as such in all front better probes for these colocalization experiments.

2. Lipidomics: The authors present a state-of-the-art lipid measurement and analysis. However, some key aspects of data acquisition, processing and visualization are missing/not presented clearly. I have listed questions below, the answers should be clearly stated somewhere in the manuscript:

a. What are the lipid standards used?

b. What are the fingerprint fragments that the lipid identification are based on?

c. The authors indicate that their method is "unbiased". However they use targeted analysis. Although they cover a large number of lipids, they are biased against the lipid species they target. Why is the approach presented as unbiased?

d. Figure S1c and d identify some of the datasets as extreme deviants. Are these datasets included in downstream analysis or excluded?

e. Within the same framework, there is a general lack of clarity. For example, I could not read the axis of supporting figures. Please provide more legible versions. Along these lines, supporting information Table legends are very difficult to access. These data tables and their legends contain key information on data processing (i.e. data normalization, statistical analysis). As such, they should be more accessible.

f. It is my understanding that due to the presence of extreme deviants (presented in Figure S1) global median centering was carried out to normalize abundances. The authors should describe how this normalization is done, provide references on the suitability of such methods for their dataset, and importantly for a given number of conditions and hypothesis tested.

g. Along these lines, I ask that the authors report raw abundances of lipids (i.e. total ion counts that for each lipid species) prior to normalization and relative abundance calculations.

h. It is also my understanding that there are two types of normalizations used i) based on protein concentrations pre-data acquisition; ii) after data acquisition to account for the "extreme variants". This should be clarified in the text when normalization is discussed.

i. How generalizable are these results in Huh7 cells? It is important to validate these findings (lipidomics and lipid perturbation experiments using inhibitors) in at least another cell line.

j. Finally, the authors have done a beautiful job with data visualization. However, in this work, these visualizations make the data and the interpretation look more complex than it actually is. One example is the correlation map in figure 2: the main conclusion of this figure is that ceramides and sphingomyelins show opposite trends, which would be much easier to demonstrate showing simply fold changes or relative abundances as a heatmap (or even in a fold-change bar plot).

3. Bioactivity of ceramides:

a. The conclusions made by authors on the involvement of specific ceramides on injection are not supported with the data presented. They present correlations. However, this is not sufficient to make such conclusive biological inferences. Additional experiments needed to perturb the levels of specific ceramides using specific CerS inactivation (just as they have done for the whole CerS family using FB1) are needed if the authors like to make these connections.

b. The changes in lipid composition (at least for de novo species) upon inhibitor treatment are studied using TLC. This provided information about the de novo changes. I am curious as to how the inhibitor treatment affect lipidome in general? This is important, because a decrease in ceramide biosynthesis by TLC does not necessarily mean that ceramide levels are decreased in the cell. After all, there are at least three major pathways that control ceramide levels. The authors should look at the overall lipid levels by LCMS in addition to TLC, and unambiguously establish overall lipid changes.

c. I strongly disagree with the statement that sphingolipids exhibit low rates of turnover. While they might be correct for some systems (and low is a relative term), it does not explain the need for a 4-day pretreatment. Both myriocin and FB1 can suppress ceramide production with much shorter treatments (i.e. 24 hours) in mammalian cells. The fact that a long pre-treatment is needed to establish the phenotype, likely mean that other downstream sphingolipids or even other lipid families that may be important for this process.

d. Ceramides are involved in numerous cellular processes including cell death. However, their role as pro-death lipids goes beyond to that of apoptotic death. For example, we now know that they are involved in other types of cell death. As such, considering apoptosis as the only potential cell death mechanism based on ceramide changes is incorrect.

e. Along these lines, the authors report some significant changes in PIs. The changes in PIs, PIPs and ceramides are lipid signatures of necroptosis, which again deemphasized pro-apoptotic lipid roles discussed.

f. Previous studies have shown associations between ceramides and other sphingolipids and viral infections. In fact, recent work on zika virus highlighted the involvement of sphingolipid biosynthesis during infection. I suggest that the authors update the literature they cite to discuss the novelty of their findings.

Response to the reviewers

We thank the reviewers for their time spent reviewing this manuscript; we found the points raised to be exceptionally insightful and thorough, and believe that our efforts to address them have greatly improved our work. Please find our point-by-point response to each comment below.

Reviewer #1 (Remarks to the Author):

In this study, Leier et al examine the changes in the lipidome during infection with Zika virus (ZIKV). They find that the largest changes are in sphingolipids, and then they provide data that increased ceramide sensitizes cells to ZIKV infection and mediates ZIKV-induced apoptosis. The lipidomics results appear robust and quite informative. The approaches to implicate ceramide/sphingomyelin also provide strong data but could be improved.

Major Points

1. What's primarily missing from this study is any insight on how ZIKV modulates sphingolipids. Does the virus effects a change in metabolism AND in subcellular localization? Are the effects on ceramide directly related to the localization effects on SM or are these two independent effects? Does ZIKV bind to SM, does it change the localization of SMS? Does it in addition inhibit ceramide incorporation into complex sphingolipids? These points should be at least discussed.

Thank you for these comments. We address each specific point below:

- *Does the virus effects a change in metabolism AND in subcellular localization?*

Yes, we have shown that cellular (revised Fig. 1) levels of both sphingomyelin (SM) and ceramide are significantly altered by ZIKV infection. Subcellular distribution of ceramide alone (revised Fig. 7) was significantly altered by ZIKV infection.

- *Are the effects on ceramide directly related to the localization effects on SM or are these two independent effects?*

One of our key findings was that cells lacking sphingomyelin synthase 1 produce 100-200-fold more infectious ZIKV than WT, strongly indicating that ceramide has a key proviral role (revised Fig. 6a). Further supporting this, we show that ceramide – but not sphingomyelin – is enriched at replication vesicle membranes. Therefore, we do not anticipate that sphingomyelin is directly required for ZIKV replication.

- *Does ZIKV bind to SM, does it change the localization of SMS? Does it in addition inhibit ceramide incorporation into complex sphingolipids?*

With superresolution microscopy, we now show that ZIKV nonstructural proteins do colocalize with ceramide at the replication complex, though this does not appear to be the case with SM (and presumably SGMS1) during infection (revised Fig. 7).

- *These points should be at least discussed.*

We have discussed these points in our revised manuscript. We have additionally updated Fig. 3a and the Discussion (paragraph 3) to include the possibility that complex sphingolipid metabolism could be involved in our findings.

2. The results in Figure 6 are very interesting results. Figure 6b plots cer/SM ratios for all species and the authors appear to suggest that this ratio is important in the regulation of ZIKV replication. It seems to this reviewer that the levels of ceramide are the ones that track best with infectious ability (more infection with high ceramide and less with less). This conclusion (if the authors agree), should be more clearly articulated. Also, did the authors measure sphingolipids, (CER and SM) after GW treatment and in SGMS1gT cells? Some kind of confirmation would be good. This is a key point that needs to be defined.

Thank you for this important comment. We have expanded and clarified our discussion of these results (revised Fig. 5 and 6) to reflect this suggestion.

We have extensively characterized SGMS1^{GT} KBM7 cells in our previous publication (Tafesse et al., 2013). In this paper, we assessed the contribution of SGMS1 to sphingolipid content using LC/MS. We find that SGMS1^{GT} cells have ~20% of total cellular SM content compared with KBM7, which corroborates ¹⁴C-choline labeling (via TLC) experiments. The decrease in SM levels in SGMS1^{GT} cells was accompanied by an increase in ceramide (see Fig. 1, Tafesse et al., 2013).

To further validate our previous data, we metabolically labeled WT and SGMS1^{GT} KBM7 cells with ¹⁴C-serine and then extracted total lipids and analyzed by TLC. As expected, these cells have a significant reduction in ¹⁴C-labelled SM as compared to the control (revised Supplementary Fig. 6a).

As suggested by the reviewer, we have now demonstrated that cells treated with GW4869 have reduced levels of SMase activity and ceramide (revised Supplementary Fig. 7).

3. The colocalization studies in Figure 3 are interesting, but they use a discordant set of probes, and the authors may be misconcluding about ceramide. Eq-SM-GFP will stain endogenous SM. C6-NBD ceramide does not stain endogenous ceramide, and it is reported and used to stain Golgi. Importantly, C6-NBD ceramide may not reflect ceramide as this can be metabolized to SM. As such, it may be staining compartments of SM formation. This should be clarified using appropriate manipulations (e.g. inhibition of SM synthesis and effects on localization of the NBD-Cer). This will affect major conclusions in Figures 3 and 5. This also affects the rationale presented on top of page

20. It actually may be even more interesting if NBC-Cer is tracking endogenous SM. That will allow improved investigation of the effects of ZIKV on SM and SMS. The authors should also note that Eq-SM-GFP may not have access to all endogenous SM, depending on exposure of SM and localization of the probe. Later the authors show many treatments that reduce SM; do these treatments reduce Eq-SM staining? PIP2 localizes differently in E vs Gag expressing cells. Is Gag driving this change in localization? Images are poor quality and often overexposed, which can mislead colocalization analysis. Importantly, the quality of the panels in Figure 3 is not optimal. The authors should also show larger fields of cells.

We thank the two referees who shared concerns (Reviewer #1, point #3 and Reviewer #3, point #1) about the design of the lipid biosensor experiments presented in Figure 3 of our initial submission, as well our choice of probes to assess sphingolipid recruitment to ZIKV replication factories (Figure 5 of the initial submission). In preparing our revised manuscript, we significantly improved upon our previous microscopy experiments, including the use of superresolution microscopy to resolve virus-lipid interactions at the ZIKV replication factory (revised Fig. 7).

We agree that our use of fluorescent lipid analogs (C6-NBD ceramide and SM-BODIPY) to visualize the distribution of intracellular sphingolipids was not sufficient to identify the lipid species that are required at ZIKV replication factories, given that the two can be interconverted and may not necessarily correlate with the distribution of endogenous lipid species. We also agree that functionalized lipids (photoactivatable/clickable) generally have comparable features with the natural cellular lipids. However, they still suffer from the fact that they wouldn't provide information regarding the dynamics of endogenous lipids during ZIKV infection, and can readily be metabolized to other sphingolipid species. To overcome this limitation, we used two approaches; 1) use of a ceramide antibody that binds to the endogenous lipid (Liu et al., 2014), and 2) a fluorescent biosensor that binds to endogenous SM (Deng et al., 2016). Using superresolution microscopy, we now show that ceramide localizes with NS4B. Visualization of SM with the lipid-binding probe Eq-SM, however, did not reveal significant colocalization with NS4B or E, supporting our lipidomics findings that ceramide is the sphingolipid category required for ZIKV replication. Eq-SM staining has been extensively validated (Deng et al., 2016, 2018; Makino et al., 2015; Yachi et al., 2012); we have also mentioned Eq-SM-accessible pools of SM in the final paragraph of the Results. In light of these new data, which largely supersede what was previously shown, we have removed our previous micrographs and placed our new images in revised Fig. 7.

Specific Points

4. The studies from Figure 1 clearly show major changes in sphingolipids that warrant further investigation based on those results; however, the presentation of Figure 2 and significance of the results are not clear to this reviewer. The authors seem to be making

a point that sphingolipids are more worthy of investigation compared to other lipids based on the results in figure 2c? but this figure is not well explained, and the rationale of what makes the broad correlations of sphingolipids more important is not clear.

Thank you for raising this concern. During revisions to the manuscript, we collected new data which provided further insight into the importance of sphingolipids in ZIKV infection (Fig. 2 of the revised manuscript) and made the original Figure 2c somewhat redundant. To incorporate both sets of data, we have replaced Figure 2 in our initial submission with these new results and moved original Figures 2a and b to revised Supplementary Fig. 2, omitting Figure 2c entirely.

5. It is not clear what the parameters were that governed the choice of diseases similar to Zika (figure 4). if the study seeks changes caused by the virus, it is not clear why these changes should be similar in other diseases only because the phenotype is 'similar'. Also, if the network analysis in figure 4 is being used as a probing experiment or as further 'fishing' for interactions of ZIKV with sphingolipids, then in that case it should be presented before Figure 3.

We thank the reviewer for bringing this point to our attention. We subjectively chose MeSH disease terms based on their similarities to different clinical characteristics of ZIKV disease, which is why we found the prominence of sphinganine and sphingosine in the metabolic network analysis so intriguing. The grouping of unrelated yet phenotypically similar pathologies into "disease modules" within the cellular interactome is a central goal of the network model of human disease. Menche et al. (2015) give a pithy description of this goal:

"If two disease modules overlap, local perturbations causing one disease can disrupt pathways of the other disease module as well, resulting in shared clinical and pathobiological characteristics... We find that disease pairs with overlapping disease modules display significant molecular similarity, elevated coexpression of their associated genes, and similar symptoms and high comorbidity. At the same time, non-overlapping disease pairs lack any detectable pathobiological relationships. The proposed network-based distance allows us to predict the pathobiological relationship even for diseases that do not share genes."

We have modified our description of the network analysis in the main text to better embody this rationale (also in Methods section). We also agree that our framing of the analysis as a 'fishing' experiment does not fit well with its placement in the manuscript. If you agree, we propose that it would best fit in as part of Fig. 6, where we examine the possible clinical implications of our findings for ZIKV disease. We have placed it there in the revised manuscript.

Why did the authors treat for 4 days with FB1 and MYR? It would appear that shorter time points may suffice. Were the SL levels studied in a time course experiment to determine the best time for treatments?

Thank you for your valuable input regarding Myriocin/FB1 treatments. In addition to the data in our initial submission showing (1) pretreatment for the time and concentrations used did not affect growth rate or morphology and (2) effectively and specifically inhibited the de novo biosynthesis pathway (revised Supplementary Fig. 5), we have performed additional lipidomics experiments on 4 day-pretreated cells, showing that levels of sphingolipids are significantly reduced in both treatments compared to vehicle, while levels of other lipids such as phospholipids are unaffected (revised Supplementary Fig. 4). To address whether the suggested 24 hr treatment time could reduce ZIKV shedding as well as the 4 day treatment, we repeated the first timepoint of the propagation assay shown in revised Fig. 3b (see below). While 24 hr FB1 pretreatment caused reduction in ZIKV replication equal to the 4 day treatment or our two SPTLC2 knockout cell lines (see Tafesse et al. (2015) for lipidomic characterization of the DC2.4 *Sptlc2*^{-/-} cell line), 24 hr myriocin treatment did not significantly reduce ZIKV replication.

Given our finding that ceramide is the key sphingolipid required for ZIKV replication, we consider the varying treatment time-dependent effects of myriocin and FB1 in light of the various ceramide fluxes in sphingolipid metabolism (reviewed by Mullen et al., 2012). An extensive salvage pathway funnels downstream sphingolipid species back to ceramide, with CerS required as a final catalytic step. FB1 therefore blocks both major routes to ceramide formation: the de novo pathway as well as the salvage pathway (revised Fig. 3a). Myriocin, on the other hand, does not prevent production of ceramides through the salvage pathway. A plausible hypothesis for the ineffectiveness of a 24 hr myriocin pretreatment is that ZIKV is able to salvage remaining downstream sphingolipids to regenerate pools of proviral ceramide even in the absence of de novo synthesis, whereas longer myriocin pretreatment – or knockout of SPTLC2 – is sufficient to deplete these salvageable metabolites. We have modified Fig. 3a, and our discussion of the inhibitors in the corresponding Results section, to reflect this broader view of sphingolipid metabolism.

Data represents plaque assays of ZIKV harvested from Huh7 cells after 48 hours post infection. Data is from three separate experiments, with technical triplicates performed in each experiment.

Also, did the authors culture these cells in serum (that would contain SM)? Perhaps this accounts for some of the observed effects on SM levels.

Yes, all of our cell culture experiments were performed in media containing 10% fetal bovine serum. As we showed that inhibition of sphingolipid metabolism greatly reduces SM levels even in the presence of serum, we expect that exogenous sphingolipids in the culture medium would have a negligible direct effect on intracellular membrane lipids.

Importantly, can exogenously applied sphingolipids rescue the effects of FB1/MYR on virus shedding? The authors appear to indicate that exogenous SM and Cer (Figure 5) can reach the virus? So do they make up for the effects of Myr/FB1?

We thank the reviewers (Reviewer #2, point 7 and Reviewer #1, point 5) who suggested performing lipid addback experiments. We performed the addback experiment following the solvent-free ceramide delivery formulation approach described previously (Kjellberg et al., 2015; Sukumaran et al., 2013). We exogenously added two types of ceramides: short-chain ceramide (C6-Cer) and long-chain ceramide (C16-Cer). Despite our repeated attempts (more than 5 times) to optimize conditions (concentration, treatment time, etc.), the addition of exogenous ceramide resulted in cell death. We were therefore not able to get a meaningful result from our experiments. While we are disappointed that we could not provide this control, we provide multiple new validations of myriocin and FB1 treatment in our revised manuscript: (1) A new genetic knockout of SPTLC2 in a human cell line (revised Fig. 3e), (2) Lipidomic analysis of myriocin and FB1-treated cells (revised Supplementary Fig. 4) showing that the effects of these inhibitors is specific to sphingolipids, (3) Experiments in other cell lines, most importantly human neural progenitor cells and neuroblastoma cells (revised Fig. 3g, h), showing treatment had the same effect on viral shedding.

6. Figure 5. The CER/SM-Bodipy correlation with ER markers are not convincing. Anyway, one has no expectation for SM to localize to the ER.

We have revised these experiments, please see our response to your major point 3.

7. While the analysis shown in Figure 7 is very nice, its conclusions are rather speculative, and is not a 'good note' to end on. Perhaps these results could be absorbed by figure 1 where changes in lipids are reported or at least moved up front?

Thank you for your insightful comment. We agree that the key findings of Figure 7 would be better placed in the context of overall Cer/SM regulation, and have therefore moved panels 7d-g of the initial submission to Fig. 5d-g of the revised manuscript.

Minor Points

1. Figure 4B. 'trivial' name is 3-ketosphinganine or 3-ketodihydrosphingosine, not 3-Dehydrosphinganine

Thank you for bringing this to our attention. The compound name has been changed to 3-ketosphinganine.

2. Not clear why the authors used 30uM myriocin. This compound is potent and specific, and it inhibits SPT at 100-500nM.

We thank the reviewer for this valuable comment. In the literature, different myriocin concentrations are reported depending on the cell type, including concentrations that are comparable to ours (Glaros et al., 2010; Mailfert et al., 2017; Orchard et al., 2018). See revised Supplementary Fig. 4 and 5 for validation that our myriocin treatment specifically reduced sphingolipid levels without affecting cell growth or viability.

3. Line 331-332; not clear; is there a decrease of 60%?

Yes, though the sentence did not make sense as written. It has been corrected.

4. The use of plotting tools is impressive, however some times excessive or not fully informative. For example, the radar plots in figure 1 don't appear to be very informative. The bubble plots don't inform which individual species are in each bubble. Perhaps some labeling of some of the more outstanding ones would be helpful. Alternatively, perhaps the authors can use a heatmap.

We agree with your assessment of the radar plots in Figure 1 and have removed them. As requested, we have labeled outstanding species in a key new bubble plot (Fig. 2d of the revised manuscript). Regarding the global bubble plots in Figure 1, we hesitate to label individual points for two reasons: (1) our intent for these panels was to present a broad overview of our lipidomics data in a visually objective manner. While we provide interpretations of the data in the main text, labelling 'outstanding' lipid species would in our view prematurely draw the attention of the reader away from broad trends in the lipidome. (2) Because bubble size and position on the y-axis have no relationship to the abundance of that species, labeling specific points may mislead the reader about the enrichment of the lipid species in intracellular membranes.

We found the bubble plot to be a more intuitive visualization of lipidome-scale trends than a heat map, but our full dataset in the Supplementary Tables includes color-coded enrichment cells that can be viewed as a heat map. Of course, the final presentations of figures are at the discretion of the referees and editor, and we will gladly make changes if asked.

5. Fig. 4d why is there virus shedding at time 0 after infection? Is this what is shown?

For many of our infection timecourses, including the one shown here, we collected a '0 hpi' set of samples immediately after removing the viral inoculum and washing away unbound virus with PBS (see Methods), well before viral shedding begins. This is a common practice in the field (Orchard et al., 2018; Zhang et al., 2018, 2016) that controls for differences between treatment groups during the inoculation itself, and establishes a level of background for the assay shown. In this case, the 0 hpi titer consists of extracellular ZIKV genomes left over from the inoculation and not removed during washing, as well as possible background from our RT-qPCR assay performed near its lower limit of detection.

Reviewer #2 (Remarks to the Author):

Thank you for asking me to review the work entitled "A global lipid map reveals a network essential for Zika virus replication" by Leier et al. In this study, the authors investigated the global lipid profile of ZIKV-infected cells. They demonstrated that ZIKV remodeled the host lipid profile after infection and that virus replication can be inhibited by modulating the ratio of the intracellular lipid species. The authors further sought to establish a link between the identified ceramide with ZIKV-induced apoptosis, which has been suggested as a contributing factor to ZIKV-induced brain/testis damage. The team has a strong record in lipid research. However, the virology of the paper is not sufficiently supported by experiments and the overall data suffers from a lack of novelty. Moreover, a significant amount of essential (raw) data were not provided and a number of details concerning the lipidomic analysis require clarifications.

Major concerns:

1. The novelty and significance of the study is limited. Lipid is already known to be required for flavivirus replication and ceramide is known as a pro-apoptotic factor. This study did not provide enough significantly new information in this aspect.

Thank you for giving us the opportunity to respond to your concern. We have included several new experiments in the revised manuscript that in our view greatly increase its novelty and strengthen our initial conclusions:

- The most extensive new experiment we performed was a lipidomic analysis of human cells ectopically expressing ZIKV NS4B, one of the non-structural proteins crucial for membrane remodeling and formation of viral replication sites (revised Fig. 2a-c). Both *Flavivirus* and Hepatitis C virus (HCV) NS4B induce membrane stress responses; HCV NS4B has been shown to play an important part in dysregulating host lipid metabolism, while the same has not been shown for flaviviruses. We found that NS4B expression, like ZIKV infection, significantly alters the lipid composition of human cells, with enrichment of sphingolipids most strongly correlated between the two conditions (revised Fig. 2d). These data not only improved the novelty and significance of our study, but also provide a possible mechanistic understanding of how ZIKV modulates the host sphingolipid network.

- This conclusion is strengthened by new superresolution microscopy experiments showing that ceramide associates with NS4B that localizes to the replication sites.
- As discussed in more detail below, we have increased the significance and relevance of our study by repeating key experiments in neural stem cells and neuroblastoma cells. We have also performed experiments in an additional genetic model of sphingolipid depletion (revised Fig. 3e).
- We have revised and expanded our discussion of the disease module-metabolic network model (revised Fig. 6), emphasizing the significance that the network perturbations we present are so over-represented in disease modules that overlap with ZIKV syndrome.

2. The analysis is not well supported by validation experiments and the presented data lacks physiological relevance. Most of the validation experiments are done in Huh7 or VeroE6 cells, which are not the most biologically relevant cells for ZIKV infection in vivo; and it is particularly problematic with Huh7 (human hepatoma) cells which has a largely changed metabolic system as a cancer cell line. Physiologically more relevant primary cell line and/or ex vivo organ culture models, and especially animal models must be applied to demonstrate the role of ceramide. More gene depletion experiments should be done to complement results from using inhibitors.

Thank you for these valuable comments. ZIKV is known to be neurotropic and mainly infects neuronal cells within the central nervous system to cause microcephaly (Mlakar et al., 2016; Oh et al., 2017). Recent studies have used iPSC-derived human neural progenitor cells for their studies as a proxy to physiologically relevant primary cells (Li et al., 2019; Muffat et al., 2018; Scaturro et al., 2018; Souza et al., 2016) Therefore, we decided to use iPSC-derived human neural progenitor cells to validate our key findings by manipulating sphingolipid levels in these cells using small molecule inhibitors. The inhibitors we used include myriocin to block serine palmitoyltransferase (SPT) and FB1 to block ceramide synthesis. We then examined if these inhibitions affect ZIKV infection by plaque assay. Inhibitor-treated cells showed a significantly reduced level of viral shading as compared to untreated cells, showing roles for sphingolipids in ZIKV infection in neuronal stem cells (Figure 3g). In addition to iPSC-derived NPCs, we also employed the human neuroblastoma cell line SH-SY5Y to validate our initial findings further. Consistent with the Huh7 and NPCs, both myriocin and FB1 significantly reduces the amount of ZIKV production (Figure 3h) as compared to the control cells. These data demonstrate roles for sphingolipid in ZIKV infection and validate our initial observations.

As suggested by the reviewer to employ gene depletion experiments in addition to inhibitors, we employed HAP1 cells that lack SPTLC2 gene, one of the subunits of SPT that is indispensable for its activity. HAP1 cells have been used in ZIKV study and to uncover host factors essential for flavivirus infection (Zhang et al., 2016). As we showed

in DC2.4 cells, we found that SPTLC2^{-/-} HAP1 cells produced significantly reduced levels of ZIKV as compared to wild type control (Figure 3e).

3. Line 119: what were the rationale and the viral replication kinetics for using 24 and 48 hpi? What were the cell conditions at the indicated time-points? Did the authors have a chance to compare lipidomic landscape under high and low MOIs?

ZIKV requires about 8-12 h to complete its life cycle (Lindenbach and Rice, 1997; Scherbik and Brinton, 2010). Since the plaque and RT-qPCR assays were performed using a low MOI (MOI of 0.1), measuring viral shading at 24, 48, and 72 h would enable us to capture the replication kinetics of ZIKV. When cells were infected with MOI of 0.1, the overall condition of the cells looks normal. We observe minor cell death at 72 h (~5%) and increased afterward. To ensure almost all the cells were infected with ZIKV, we used higher MOI for the lipidomic study.

4. Figure 2b and Figure 7c: the titles are missing in the X-axis.

Thank you for catching this omission. The x-axis of the two figures are now labeled "Lipid species".

5. Figure 3. It is not appropriate to use HIV Gag as a control for ZIKV prM+E. HIV Gag+Env should be used in this case. If immunostaining is against ZIKV E, the counter staining should be on HIV Env.

In preparing our revised manuscript, we significantly expanded upon our previous microscopy experiments mapping the distribution of sphingomyelin and ceramide within infected cells, including using superresolution microscopy to resolve virus-lipid interactions at the ZIKV replication complex. In light of these new data, which largely supersede what is shown here, we have decided to remove Figure 3 from the revised manuscript. Please see our responses to Reviewer #1, point 3 and Reviewer #3, point 1.

6. Figure 4f: explain why in Sptlc2^{-/-} cells, the ZIKV virus titer in 24hpi was even lower than that of the baseline at 0 hpi? It would be important to show the data at later time-points like 48 and 72hpi.

DC2.4 is a murine dendritic cell line that in our hands is highly resistant to flavivirus infection. By 24 hpi, *Sptlc2*^{-/-} cells appear to restrict ZIKV replication to the point where viral shedding falls below the background levels seen immediately after inoculation and washing. Our initial experiments included 48 and 72 hpi samples, but we found that viral levels in the supernatant of both WT and *Sptlc2*^{-/-} cells was very low at later timepoints, indicating that DC2.4 cells were largely able to clear the infection. In preparing the revised manuscript, we repeated this experiment in permissive HAP1 human cells lacking SPTLC2, where we observed increases in infection over time (Figure 3e). Though DC2.4 is clearly not an ideal model for ZIKV infection, we wish to keep this data in the manuscript (now in Figure 3f, with 0 hpi removed to avoid confusion) to show that

the phenotype we identified is not dependent on cell type or restriction by innate immunity.

7. Figure 4b: ZIKV rescue assays must be performed to determine if addition of supplementary ceramide or other products along the pathway can reverse the anti-ZIKV effect of Myr and/or FB1.

We thank the reviewers (Reviewer #2, point 7 and Reviewer #1, point 5) who suggested performing lipid addback experiments. We performed the addback experiment following the solvent-free ceramide delivery formulation approach described previously (Kjellberg et al., 2015; Sukumaran et al., 2013). We exogenously added two types of ceramides: short-chain ceramide (C6-Cer) and long-chain ceramide (C16-Cer). Despite our repeated attempts (more than 5 times) to optimize conditions (concentration, treatment time, etc.), the addition of exogenous ceramide resulted in cell death. We were therefore not able to get a meaningful result from our experiments. While we are disappointed that we could not provide this control, we provide multiple new validations of myriocin and FB1 treatment in our revised manuscript: (1) A new genetic knockout of SPTLC2 in a human cell line (revised Fig. 3e), (2) Lipidomic analysis of myriocin and FB1-treated cells (revised Supplementary Fig. 4) showing that the effects of these inhibitors is specific to sphingolipids, (3) Experiments in other cell lines, most importantly human neural progenitor cells and neuroblastoma cells (revised Fig. 3g, h), showing treatment had the same effect on viral shedding.

8. Sample collection:

- Why were large dishes (15cm) used to culture low intensity cell (1.2×10^6 cells) – what are the advantages of this procedure? More detailed description about the protein normalization procedures must be provided. The normalization is an essential step with high MOI infection (with an MOI > 50 PFU/cell) and the infected cells should be dead at 24 and 48hrs. So when the cells were collected, the cell number must be very different between mock and infected groups, which will strongly influence the subsequent selection of significantly perturbed lipids.

Our principal concern in performing the 24-48 hpi time-course experiment introduced in Figure 1 was to limit differences in lipid metabolism between mock and infected samples to those induced by ZIKV replication. A potentially confounding effect we encountered was the tendency of ZIKV-infected cells to divide at a much slower rate than mock cells, raising the possibility of changes in mock cell lipid metabolism caused by overcrowding at 48 hpi. Because Huh7 cells are quite large, we erred on the side of caution in using 15-cm dishes and indeed found that mock cells grew to 70-80% confluency by the end of the experiment. We did not observe significant cytopathic effects in infected cells at 24 or 48 hpi, and the difference in density between mock and infected cells actually decreased from 24 and 48 hpi

9. Lipid extraction:

- The lower organic lipid containing layer was removed, dried in vacuum. Why was it stored at -20°C in 2:1 chloroform:methanol (v/v) until analysis (line 587)? Lipids extract was dried in vacuo and reconstituted in 10 µl chloroform plus 540 µl methanol (line 591). Why was it necessary to perform storage in 2:1 chloroform:methanol (v/v) if the final reconstituted solvent was 10 µl chloroform plus 540 µl methanol?

When total lipid extracts (TLEs) are stored at -20°C in methanol, we have noticed some of the lipids may fall out of solution (e.g. PE lipids). To keep all lipids reconstituted during storage, we store the TLEs in 2:1 chloroform/methanol until they are ready for MS analysis.

10. LC-MS/MS analysis and lipid identification:

- Were the corresponding internal standards included during the sample preparation and LC-MS analysis? These could not be found in the main text.

Our lipidomic analysis was performed using Orbitrap mass spectrometer that is based on exact mass/fragmentation identification. Internal standards were not used for this study. To facilitate quantification of lipids, a reference database for lipids identified from the MS/MS data was created and features from each analysis were then aligned to the reference database based on their identification, m/z and retention time using MZmine 2. Aligned features were manually verified and peak apex intensity values were exported for subsequent statistical analysis.

The approach used in this manuscript, label-free quantitation, has been employed for the last two decades in LC-MS-based proteomics and is accurate in identifying relative abundance differences in detected peptides. Therefore, the same approach is reasonable in global LC-MS-based lipidomics experiments. Indeed we and others have utilized the label-free relative quantification approach in several peer-reviewed publications and shown it to be successful at identifying relative differences in lipid species in global LC-MS-based lipidomics experiments (Dautel et al., 2017; Diamond et al., 2010; Eisfeld et al., 2017; Kyle et al., 2018, 2019; Perera et al., 2012; Sorensen et al., 2010; t'Kindt et al., 2015; Telenga et al., 2014; Tisoncik-Go et al., 2016). We have made sure the Methods section of the manuscript describes label-free relative quantification.

- The authors must provide the table of the 340 identified lipids.

We apologize for formatting errors that made this table inaccessible in our initial submission. The table of 340 identified lipids can be found in Supplementary Data 1 of the revised manuscript.

10. Supplementary data1. Total protein content of cell lysate used for lipidomics: the protein concentration data in different cell samples could not be found. The authors must show these data and demonstrate how was the protein concentration used to normalize the MS intensity of individual sample (if total intensity normalization was used)?

Protein content was measured to approximate differences in cell density between mock and infected samples, not for normalization during MS. Our normalization method is detailed in the Methods section of the manuscript, see also our response above. We apologize for the confusion.

11. Supplementary data2. The authors mentioned that 340 lipids were identified and these lipids' changes in normalized abundance between mock and infected cells could be calculated (line 118 to 120 in main text). However, these corresponding data were not found in the supplemental data 2 - only the column header and descriptions were shown.

We apologize for the poor state of the supplementary tables in our initial submission file, which was due to errors in formatting during the submission process. We have corrected the errors in our resubmitted Supplementary Data.

12. Supplementary data3. The significant lipids and corresponding information such as ionization mode, associated adduct, m/z, retention time (RT), p-values, and log 2 fold change (FC) could not be observed in the supplemental data 3.

We have corrected the errors in our resubmitted Supplementary Data.

13. Supplementary data 5. The raw data of calculated ratios of normalized lipid levels for all possible pairs of SM and Cer species also could not be found in the Supplementary Data 5. The raw lipids table containing the calculated ratios values and statistical values must be shown.

We have corrected the errors in our resubmitted Supplementary Data.

Reviewer #3 (Remarks to the Author):

Leier et al analyzed the changes in lipid composition after injection with ZIKV. They identified sphingolipids as an important mediator of virulence and provided information on the role of sphingolipids in viral replication.

My enthusiasm on the state-of-the-art lipid measurements are hampered by important details needing clarification and the use of fluorescently labeled lipid to study lipid localization. Please find my questions and recommendations on these below:

1.Lipid localization: Previous studies have already linked sphingolipids to ZIKV infection, as such, the novelty of the work mostly lies on the lipid colocalization with viral replication. Although fluorescent lipid analogs can be useful in studying more dynamic questions, the colocalization experiment can be (and in my opinion should be) carried out using probes with less perturbation on lipid structure such as clickable lipids. These

lipids (alkynyl versions) are commercially available, can be used in fixed and live cells and exhibit a strong structural similarity to the endogenous lipids, as such in all front better probes for these colocalization experiments.

We thank the two referees who shared concerns (Reviewer #1, point #3 and Reviewer #3, point #1) about the design of the lipid biosensor experiments presented in Figure 3 of our initial submission, as well our choice of probes to assess sphingolipid recruitment to ZIKV replication factories (Figure 5 of the initial submission). In preparing our revised manuscript, we significantly improved upon our previous microscopy experiments, including the use of superresolution microscopy to resolve virus-lipid interactions at the ZIKV replication factory (revised Fig. 7).

We agree that our use of fluorescent lipid analogs (C6-NBD ceramide and SM-BODIPY) to visualize the distribution of intracellular sphingolipids was not sufficient to identify the lipid species that are required at ZIKV replication factories, given that the two can be interconverted and may not necessarily correlate with the distribution of endogenous lipid species. We also agree that functionalized lipids (photoactivatable/clickable) generally have comparable features with the natural cellular lipids. However, they still suffer from the fact that they wouldn't provide information regarding the dynamics of endogenous lipids during ZIKV infection, and can readily be metabolized to other sphingolipid species. To overcome this limitation, we used two approaches; 1) use of a ceramide antibody that binds to the endogenous lipid (Liu et al., 2014), and 2) a fluorescent biosensor that binds to endogenous SM (Deng et al., 2016). Using superresolution microscopy, we now show that ceramide localizes with NS4B. Visualization of SM with the lipid-binding probe Eqt-SM, however, did not reveal significant colocalization with NS4B or E, supporting our lipidomics findings that ceramide is the sphingolipid category required for ZIKV replication. We have revised the text to reflect these data. In light of these new data, which largely supersede what was previously shown, we have removed our previous micrographs and placed our new images in revised Fig. 7.

2. Lipidomics: The authors present a state-of-the-art lipid measurement and analysis. However, some key aspects of data acquisition, processing and visualization are missing/not presented clearly. I have listed questions below, the answers should be clearly stated somewhere in the manuscript:

a. What are the lipid standards used?

Our lipidomic analysis was performed using Orbitrap mass spectrometer that is based on exact mass/fragmentation identification. Internal standards were not used for this study. To facilitate quantification of lipids, a reference database for lipids identified from the MS/MS data was created and features from each analysis were then aligned to the reference database based on their identification, m/z and retention time using MZmine 2. Aligned features were manually verified and peak apex intensity values were

exported for subsequent statistical analysis. Please also see our response to Reviewer #2 point 10.

b. What are the fingerprint fragments that the lipid identification are based on?

The fragment ions used for lipid identifications were used as outlined in Kyle et al. (2017). Briefly, fragments ions corresponding to the the diagnostic ions (if applicable) and the corresponding chain fragments were utilized. In the example below, we are showing the lipid PC(16:0/18:1), were the red fragment is diagnostic ion (m/z 184), and the green fragments are the matching chain fragments to that specific identification (all the observed fragments that match the identification are shown in the 'Observed MS/MS table). All fragments that do not match the identification are gray, and as you can see, are very minor in the example. For this example, and all of our diacyl-PC lipids, we use the diagnostic fragments in the image (m/z 184 but also 104, 125) as well the fatty acid fragments (M-fatty acid (for both chains), M-ketene (for both chains), and DAG (or M-head group). We have added this information in the Method Section.

c. The authors indicate that their method is “unbiased”. However they use targeted analysis. Although they cover a large number of lipids, they are biased against the lipid species they target. Why is the approached presented as unbiased?

You are correct. When making our identifications, in positive (POS) mode we have a list of over 14,000 lipids in the database (see “# targets: 14089 in the above figure) and over 8700 in negative (NEG) mode, and if a lipid is not in that database we may not identify that particular lipid. Our databases are sourced from Lipid Maps (primarily a mammalian based lipid database) and modified by the user (Jennifer Kyle, in this case) such that new lipid species, once identified, can be added to the database. Our meaning of unbiased was based on this and our attempt to highlight that our approach was not targeted, which may be interpreted as selected reaction monitoring (SRM) type of analyses. We nonetheless agree that our usage of the term may be misleading and have removed it from the manuscript wherever it does not have its technical definition. Thank you for pointing out this important distinction.

d. Figure S1c and d identify some of the datasets as extreme deviants. Are these datasets included in downstream analysis or excluded?

They were excluded, as we now state in the legend for revised Supplementary Fig. 1.

e. Within the same framework, there is a general lack of clarity. For example, I could not read the axis of supporting figures. Please provide more legible versions. Along these lines, supporting information Table legends are very difficult to access. These data tables and their legends contain key information on data processing (i.e. data normalization, statistical analysis). As such, they should be more accessible.

We apologize for changes in formatting during the submission process that lowered the quality of the figures and made parts of the Supplementary Data inaccessible. We have corrected these errors in our resubmission.

f. It is my understanding that due to the presence of extreme deviants (presented in Figure S1) global median centering was carried out to normalize abundances. The authors should describe how this normalization is done, provide references on the suitability of such methods for their dataset, and importantly for a given number of conditions and hypothesis tested.

Global median centering, as described in (Callister et al., 2006; Wang et al., 2006; Yang et al., 2002) is a standard approach for normalizing both MS lipidomics and metabolomics data, where missing values are minimal (e.g. in comparison to MS proteomics data, which is the focus of some of these articles). This approach has been utilized in numerous publications, and we note a selection of them here (Dautel et al., 2017; De Livera et al., 2012; Polpitiya et al., 2008). We have included this information in the Methods.

g. Along these lines, I ask that the authors report raw abundances of lipids (i.e. total ion counts that for each lipid species) prior to normalization and relative abundance calculations.

We have provided the peak apex intensities for each lipidomics experiment in the Source Data file.

h. it is also my understanding that there are two types of normalizations used i) based on protein concentrations pre-data acquisition; ii) after data acquisition to account for the “extreme variants”. This should be clarified in the text when normalization is discussed.

We apologize for the confusion. We did not use protein concentrations for normalization. Please refer to the *QC, normalization, and statistical comparison methods* section in the Methods.

i. How generalizable are these results in Huh7 cells? It is important to validate these findings (lipidomics and lipid perturbation experiments using inhibitors) in at least another cell line.

Thank you for these valuable comments. Please see our response to Reviewer #2 point 2.

j. Finally, the authors have done a beautiful job with data visualization. However, in this work, these visualizations make the data and the interpretation look more complex than it actually is. One example is the correlation map in figure 2: the main conclusion of this figure is that ceramides and sphingomyelins show opposite trends, which would be much easier to demonstrate showing simply fold changes or relative abundances as a heatmap (or even in a fold-change bar plot).

While we share the reviewer’s enthusiasm for data visualization, we regrettably agree that in places this came at the cost of clarity for the reader. We have made the following changes to simplify our presentation:

- Fig. 1d and f (initial submission) were deemed unnecessary and removed.
- Fig. 2c (initial submission) was removed and panels a-b were moved to Supplementary Fig. 2, where they are now integrated into our discussion of Fig. 1.
- Some of our lipidomics data is now presented as fold-change bar plots (see Supplementary Fig. 4).

3. Bioactivity of ceramides:

a. The conclusions made by authors on the involvement of specific ceramides on injection are not supported with the data presented. They present correlations. However, this is not sufficient to make such conclusive biological inferences. Additional experiments needed to perturb the levels of specific ceramides using specific CerS inactivation (just as they have done for the whole CerS family using FB1) are needed if the authors like to make these connections.

Thank you for sharing our appreciation of the novel and potentially significant finding that ZIKV infection dysregulates ceramide metabolism on the level of acyl chain identity,

which is now further supported by lipidomics of NS4B-expressing cells (see revised Fig. 2). We agree, however, that to support some of the connections we make on the basis of this finding would require extensive validation experiments beyond the scope of this manuscript, and have removed them from our discussion of these results.

b. The changes in lipid composition (at least for de novo species) upon inhibitor treatment are studied using TLC. This provided information about the de novo changes. I am curious as to how the inhibitor treatment affect lipidome in general? This is important, because a decrease in ceramides biosynthesis by TLC does not necessarily mean that ceramide levels are decreased in the cell. After all, there are at least three major pathways that control ceramide levels. The authors should look at the overall lipid levels by LCMS in addition to TLC, and unambiguously establish overall lipid changes.

We thank the reviewer for this suggestion. We have now performed lipid profiling of cells that were treated with myriocin and FB1 for 4 days (Supplementary Fig. 4) and found, as shown by TLC, that sphingolipid levels were reduced while phospholipids were not affected.

c. I strongly disagree with the statement that sphingolipids exhibit low rates of turnover. While they might be correct for some systems (and low is a relative term), it does not explain the need for a 4-day pretreatment. Both myriocin and FB1 can suppress ceramide production with much shorter treatments (i.e. 24 hours) in mammalian cells. The fact that a long pre-treatment is needed to establish the phenotype, likely mean that other downstream sphingolipids or even other lipid families that may be important for this process.

Thank you for your valuable input regarding Myriocin/FB1 treatments. In addition to the data in our initial submission showing (1) pretreatment for the time and concentrations used did not affect growth rate or morphology and (2) effectively and specifically inhibited the de novo biosynthesis pathway (revised Supplementary Fig. 5), we have performed additional lipidomics experiments on 4 day-pretreated cells, showing that levels of sphingolipids are significantly reduced in both treatments compared to vehicle, while levels of other lipids such as phospholipids are unaffected (revised Supplementary Fig. 4). To address whether the suggested 24 hr treatment time could reduce ZIKV shedding as well as the 4 day treatment, we repeated the first timepoint of the propagation assay shown in revised Fig. 3b (see below). While 24 hr FB1 pretreatment caused reduction in ZIKV replication equal to the 4 day treatment or our two SPTLC2 knockout cell lines (see Tafesse et al. (2015) for lipidomic characterization of the DC2.4 *Sptlc2*^{-/-} cell line), 24 hr myriocin treatment did not significantly reduce ZIKV replication.

Given our finding that ceramide is the key sphingolipid required for ZIKV replication, we consider the varying treatment time-dependent effects of myriocin and FB1 in light of the various ceramide fluxes in sphingolipid metabolism (reviewed by Mullen et al., 2012). An extensive salvage pathway funnels downstream sphingolipid species back to ceramide, with CerS required as a final catalytic step. FB1 therefore blocks both major routes to ceramide formation: the de novo pathway as well as the salvage pathway

(revised Fig. 3a). Myriocin, on the other hand, does not prevent production of ceramides through the salvage pathway. A plausible hypothesis for the ineffectiveness of a 24 hr myriocin pretreatment is that ZIKV is able to salvage remaining downstream sphingolipids to regenerate pools of proviral ceramide even in the absence of de novo synthesis, whereas longer myriocin pretreatment – or knockout of SPTLC2 – is sufficient to deplete these salvageable metabolites. We have modified Fig. 3a, and our discussion of the inhibitors in the corresponding Results section, to reflect this broader view of sphingolipid metabolism.

Data represents plaque assays of ZIKV harvested from Huh7 cells after 48 hours post infection. Data is from three separate experiments, with technical triplicates performed in each experiment.

d. Ceramides are involved in numerous cellular processes including cell death. However, their role as pro-death lipids goes beyond to that of apoptotic death. For example, we now know that they are involved in other types of cell death. As such, considering apoptosis as the only potential cell death mechanism based on ceramide changes is incorrect.

e. Along these lines, the authors report some significant changes in PIs. The changes in PIs, PIPs and ceramides are lipid signatures of necroptosis, which again deemphasized pro-apoptotic lipid roles discussed.

Thank you for emphasizing the need to carefully distinguish between the many interactions that diverse lipid classes have with cell death pathways (Green et al., 2014). As we responded to an earlier comment, rigorously demonstrating a role for specific lipids in flavivirus-induced cell death is an important question deserving of extensive experiments that we feel are beyond the scope of this manuscript. We have altered our discussion of cell death to focus on associations in the literature without making causal claims from our data.

f. Previous studies have shown associations between ceramides and other sphingolipids and viral infections. In fact, recent work on zika virus highlighted the

involvement of sphingolipid biosynthesis during infection. I suggest that the authors update the literature they cite to discuss the novelty of their findings.

We have cited all relevant literature and updated our reference list.

Letter-only references

- Callister, S.J., Barry, R.C., Adkins, J.N., Johnson, E.T., Qian, W.-J., Webb-Robertson, B.-J.M., Smith, R.D., and Lipton, M.S. (2006). Normalization approaches for removing systematic biases associated with mass spectrometry and label-free proteomics. *J. Proteome Res.* 5, 277–286.
- Dautel, S.E., Kyle, J.E., Clair, G., Sontag, R.L., Weitz, K.K., Shukla, A.K., Nguyen, S.N., Kim, Y.-M., Zink, E.M., Luders, T., et al. (2017). Lipidomics reveals dramatic lipid compositional changes in the maturing postnatal lung. *Sci. Rep.* 7, 40555.
- Deng, Y., Rivera-Molina, F.E., Toomre, D.K., and Burd, C.G. (2016). Sphingomyelin is sorted at the trans Golgi network into a distinct class of secretory vesicle. *Proc Natl Acad Sci U S A* 113, 6677–6682.
- Deng, Y., Pakdel, M., Blank, B., Sundberg, E.L., Burd, C.G., and von Blume, J. (2018). Activity of the SPCA1 Calcium Pump Couples Sphingomyelin Synthesis to Sorting of Secretory Proteins in the Trans-Golgi Network. *Dev. Cell* 47, 464-478.e8.
- Diamond, D.L., Syder, A.J., Jacobs, J.M., Sorensen, C.M., Walters, K.-A., Proll, S.C., McDermott, J.E., Gritsenko, M.A., Zhang, Q., Zhao, R., et al. (2010). Temporal proteome and lipidome profiles reveal hepatitis C virus-associated reprogramming of hepatocellular metabolism and bioenergetics. *PLoS Pathog.* 6, e1000719.
- Eisfeld, A.J., Halfmann, P.J., Wendler, J.P., Kyle, J.E., Burnum-Johnson, K.E., Peralta, Z., Maemura, T., Walters, K.B., Watanabe, T., Fukuyama, S., et al. (2017). Multi-platform 'Omics Analysis of Human Ebola Virus Disease Pathogenesis. *Cell Host Microbe* 22, 817-829.e8.
- Glaros, E.N., Kim, W.S., and Garner, B. (2010). Myriocin-mediated up-regulation of hepatocyte apoA-I synthesis is associated with ERK inhibition. *Clin. Sci. (Lond.)* 118, 727–736.
- Green, D.R., Galluzzi, L., and Kroemer, G. (2014). Metabolic control of cell death. *Science* (80-.). 345, 1250256.
- Kjellberg, M.A., Lönnfors, M., Slotte, J.P., and Mattjus, P. (2015). Metabolic conversion of ceramides in HeLa cells - A cholesteryl phosphocholine delivery approach. *PLoS One* 10, 1–19.
- Kyle, J.E., Clair, G., Bandyopadhyay, G., Misra, R.S., Zink, E.M., Bloodsworth, K.J., Shukla, A.K., Du, Y., Lillis, J., Myers, J.R., et al. (2018). Cell type-resolved human lung lipidome reveals cellular cooperation in lung function. *Sci. Rep.* 8, 13455.
- Kyle, J.E., Burnum-Johnson, K.E., Wendler, J.P., Eisfeld, A.J., Halfmann, P.J., Watanabe, T., Sahr, F., Smith, R.D., Kawaoka, Y., Waters, K.M., et al. (2019). Plasma lipidome reveals critical illness and recovery from human Ebola virus disease. *Proc Natl Acad Sci U S A* 116, 3919–3928.
- Li, Y., Muffat, J., Omer Javed, A., Keys, H.R., Lungjangwa, T., Bosch, I., Khan, M., Virgilio, M.C., Gehrke, L., Sabatini, D.M., et al. (2019). Genome-wide CRISPR screen for Zika virus resistance in human neural cells. *Proc Natl Acad Sci U S A* 201900867.
- Lindenbach, B.D., and Rice, C.M. (1997). trans-Complementation of yellow fever virus NS1 reveals a role in early RNA replication. *J. Virol.* 71, 9608–9617.
- Liu, Y., Samuel, B.S., Breen, P.C., and Ruvkun, G. (2014). *Caenorhabditis elegans* pathways that surveil and defend mitochondria. *Nature* 508, 406–410.

- De Livera, A.M., Dias, D.A., De Souza, D., Rupasinghe, T., Pyke, J., Tull, D., Roessner, U., McConville, M., and Speed, T.P. (2012). Normalizing and integrating metabolomics data. *Anal. Chem.* *84*, 10768–10776.
- Mailfert, S., Hamon, Y., Bertaux, N., He, H.-T., and Marguet, D. (2017). A user's guide for characterizing plasma membrane subdomains in living cells by spot variation fluorescence correlation spectroscopy. *Methods Cell Biol.* *139*, 1–22.
- Makino, A., Abe, M., Murate, M., Inaba, T., Yilmaz, N., Hullin-Matsuda, F., Kishimoto, T., Schieber, N.L., Taguchi, T., Arai, H., et al. (2015). Visualization of the heterogeneous membrane distribution of sphingomyelin associated with cytokinesis, cell polarity, and sphingolipidosis. *FASEB J.* *29*, 477–493.
- Menche, J., Sharma, A., Kitsak, M., Ghiassian, S.D., Vidal, M., Loscalzo, J., and Barabási, A.-L. (2015). Uncovering disease-disease relationships through the incomplete interactome. *Science* (80-). *347*, 1257601.
- Mlakar, J., Korva, M., Tul, N., Popović, M., Poljšak-Prijatelj, M., Mraz, J., Kolenc, M., Rus, K.R., Vipotnik, T.V., Vodusek, V.F., et al. (2016). Zika virus associated with microcephaly. *N. Engl. J. Med.* *374*, 951–958.
- Muffat, J., Li, Y., Omer, A., Durbin, A., Bosch, I., Bakiasi, G., Richards, E., Meyer, A., Gehrke, L., and Jaenisch, R. (2018). Human induced pluripotent stem cell-derived glial cells and neural progenitors display divergent responses to Zika and dengue infections. *Proc Natl Acad Sci U S A* *115*, 7117–7122.
- Mullen, T.D., Hannun, Y.A., and Obeid, L.M. (2012). Ceramide synthases at the centre of sphingolipid metabolism and biology. *Biochem. J.* *441*, 789–802.
- Oh, Y., Zhang, F., Wang, Y., Lee, E.M., Choi, I.Y., Lim, H., Mirakhori, F., Li, R., Huang, L., Xu, T., et al. (2017). Zika virus directly infects peripheral neurons and induces cell death. *Nat Neurosci* *20*, 1209–1212.
- Orchard, R.C., Wilen, C.B., and Virgin, H.W. (2018). Sphingolipid biosynthesis induces a conformational change in the murine norovirus receptor and facilitates viral infection. *Nat. Microbiol.* *3*, 1109–1114.
- Perera, R., Riley, C., Isaac, G., Hopf-Jannasch, A.S., Moore, R.J., Weitz, K.W., Pasa-Tolic, L., Metz, T.O., Adamec, J., and Kuhn, R.J. (2012). Dengue virus infection perturbs lipid homeostasis in infected mosquito cells. *PLoS Pathog* *8*, e1002584.
- Polpitiya, A.D., Qian, W.-J., Jaitly, N., Petyuk, V.A., Adkins, J.N., Camp, D.G. 2nd, Anderson, G.A., and Smith, R.D. (2008). DAnTE: a statistical tool for quantitative analysis of -omics data. *Bioinformatics* *24*, 1556–1558.
- Scaturro, P., Stukalov, A., Haas, D.A., Cortese, M., Draganova, K., Płaszczycyca, A., Bartenschlager, R., Götz, M., and Pichlmair, A. (2018). An orthogonal proteomic survey uncovers novel Zika virus host factors. *Nature* *561*, 253–257.
- Scherbik, S. V, and Brinton, M.A. (2010). Virus-induced Ca²⁺ influx extends survival of west nile virus-infected cells. *J. Virol.* *84*, 8721–8731.
- Sorensen, C.M., Ding, J., Zhang, Q., Alquier, T., Zhao, R., Mueller, P.W., Smith, R.D., and Metz, T.O. (2010). Perturbations in the lipid profile of individuals with newly diagnosed type 1 diabetes mellitus: lipidomics analysis of a Diabetes Antibody Standardization Program sample subset. *Clin. Biochem.* *43*, 948–956.
- Souza, B.S.F., Sampaio, G.L.A., Pereira, C.S., Campos, G.S., Sardi, S.I., Freitas, L.A.R., Figueira, C.P., Paredes, B.D., Nonaka, C.K. V., Azevedo, C.M., et al. (2016). Zika virus infection induces mitosis abnormalities and apoptotic cell death of human

- neural progenitor cells. *Sci. Rep.* 6, 39775.
- Sukumaran, P., Lönnfors, M., Långvik, O., Pulli, I., Törnquist, K., and Slotte, J.P. (2013). Complexation of C6-Ceramide with Cholesteryl Phosphocholine - A Potent Solvent-Free Ceramide Delivery Formulation for Cells in Culture. *PLoS One* 8.
- t'Kindt, R., Telenga, E.D., Jorge, L., Van Oosterhout, A.J.M., Sandra, P., Ten Hacken, N.H.T., and Sandra, K. (2015). Profiling over 1500 lipids in induced lung sputum and the implications in studying lung diseases. *Anal. Chem.* 87, 4957–4964.
- Tafesse, F.G., Sanyal, S., Ashour, J., Guimaraes, C.P., Hermansson, M., Somerharju, P., and Ploegh, H.L. (2013). Intact sphingomyelin biosynthetic pathway is essential for intracellular transport of influenza virus glycoproteins. *Proc Natl Acad Sci U S A* 110, 6406–6411.
- Telenga, E.D., Hoffmann, R.F., t'Kindt, R., Hoonhorst, S.J.M., Willemse, B.W.M., van Oosterhout, A.J.M., Heijink, I.H., van den Berge, M., Jorge, L., Sandra, P., et al. (2014). Untargeted lipidomic analysis in chronic obstructive pulmonary disease. Uncovering sphingolipids. *Am. J. Respir. Crit. Care Med.* 190, 155–164.
- Tisoncik-Go, J., Gasper, D.J., Kyle, J.E., Eisfeld, A.J., Selinger, C., Hatta, M., Morrison, J., Korth, M.J., Zink, E.M., Kim, Y.-M., et al. (2016). Integrated Omics Analysis of Pathogenic Host Responses during Pandemic H1N1 Influenza Virus Infection: The Crucial Role of Lipid Metabolism. *Cell Host Microbe* 19, 254–266.
- Wang, P., Tang, H., Zhang, H., Whiteaker, J., Paulovich, A.G., and Mcintosh, M. (2006). Normalization regarding non-random missing values in high-throughput mass spectrometry data. *Pac. Symp. Biocomput.* 315–326.
- Yachi, R., Uchida, Y., Balakrishna, B.H., Anderluh, G., Kobayashi, T., Taguchi, T., and Arai, H. (2012). Subcellular localization of sphingomyelin revealed by two toxin-based probes in mammalian cells. *Genes to Cells* 17, 720–727.
- Yang, Y.H., Dudoit, S., Luu, P., Lin, D.M., Peng, V., Ngai, J., and Speed, T.P. (2002). Normalization for cDNA microarray data: a robust composite method addressing single and multiple slide systematic variation. *Nucleic Acids Res.* 30, e15.
- Zhang, J., Lan, Y., Li, M.Y., Lamers, M.M., Fusade-Boyer, M., Klemm, E., Thiele, C., Ashour, J., and Sanyal, S. (2018). Flaviviruses Exploit the Lipid Droplet Protein AUP1 to Trigger Lipophagy and Drive Virus Production. *Cell Host Microbe* 23, 819-831.e5.
- Zhang, R., Miner, J.J., Gorman, M.J., Rausch, K., Ramage, H., White, J.P., Zuiani, A., Zhang, P., Fernandez, E., Zhang, Q., et al. (2016). A CRISPR screen defines a signal peptide processing pathway required by flaviviruses. *Nature* 535, 164–168.

Reviewers' Comments:

Reviewer #2:

Remarks to the Author:

Dear Editor Dr. Schmid,

Thank you for asking me to review the revised version of "A global lipid map defines a network essential for Zika virus replication" by Tafesse et al.

While the authors have performed additional experiments and clarified most of the previous comments about the lipidomics analysis, the virological experiments are not substantial enough to support their proposed claims, especially those related to their claimed novelty of the study. Below are the major issues that should be thoroughly addressed before publication in your prestigious journal should be considered:

1. More mechanistic investigations should be done on how exactly ZIKV modulates the sphingolipid network.
2. How are sphingolipids required for ZIKV replication? Figure 7 shows colocalization between NS4B and ceramide but this finding is preliminary. Importantly, is the requirement on sphingolipid network and ceramide for ZIKV replication really specific? Other lipids should be added as controls. Will ZIKV replication be compromised if other lipid biosynthesis pathways are perturbed? Will NS4B colocalized with other lipids? There are insufficient controls and pathway analysis done to ascertain their claims!
3. Since the ceramide supplement experiment was not successful, other experiments must be done to demonstrate the physiological importance of the sphingolipid network as also suggested by the authors. This MUST be performed with KO mice and/or inhibitors in vivo.
4. In cell-based lipidomics or metabolomics, the cell number is essential and would significantly affect the MS data analysis. The cell counting, DNA and protein quantification in different groups should be used to normalize the MS data for appropriate statistical analyses. Although the authors indicated that no significant cytopathic effects were observed and that the measured protein showed no differences, the protein quantification results should be provided to unambiguously show the lack of differences between the mock-infected and virus-infected groups.
5. Lipids standards / internal controls should be shown for the lipid extraction to assess the variance of the individual lipid species in different samples' extractions, the extracted lipid efficiency and coverage for the label-free relative quantification approach in global LC-MS-based lipidomics experiments.

Reviewer #3:

Remarks to the Author:

The authors have addressed my questions and comments sufficiently. Thank you.

Ekin Atilla-Gokcumen

Reviewer #4:

Remarks to the Author:

I have gone through the report of referee 1 and the responses to this referee. I find the responses satisfactory. I did not go into the details of the lipidomics data pipeline because this was also addressed by referee 3 who will also get a chance to look at the revision. I do not understand why the authors chose to ignore the changes in PC and PI (even PC/PE ratios could be important). When they analyse their data they look mainly at fold changes and therefore, the changes in ceramide look stronger than for PC and PI, but PC and PI are more abundant lipids, so it is more difficult to get large fold changes.

Response to the reviewers

We thank the reviewers for their positive reception of our revised manuscript; in addressing your comments and questions, we believe the manuscript has been strengthened. In response to additional points raised in our resubmission, we have performed new sets of experiments to address these concerns and modified the manuscript accordingly. We prepared point-by-point responses to each comment on our revisions below.

Reviewers' comments:

Reviewer #2 (Remarks to the Author):

Dear Editor Dr. Schmid,

Thank you for asking me to review the revised version of “A global lipid map defines a network essential for Zika virus replication” by Tafesse et al.

While the authors have performed additional experiments and clarified most of the previous comments about the lipidomics analysis, the virological experiments are not substantial enough to support their proposed claims, especially those related to their claimed novelty of the study. Below are the major issues that should be thoroughly addressed before publication in your prestigious journal should be considered:

1. 1. More mechanistic investigations should be done on how exactly ZIKV modulates the sphingolipid network.

We thank the reviewer for their comment. In Fig. 2 we demonstrate a mechanism by which NS4B can directly regulate the sphingolipid network independent of other viral proteins. We have now expanded our discussion in the text (Lines 147-179) to include literature references relevant to this mechanism. While fully elucidating the molecular underpinnings of NS4B regulation will involve biochemical assays beyond the scope of this paper, we very much plan to pursue these studies in the future.

2. How are sphingolipids required for ZIKV replication? Figure 7 shows colocalization between NS4B and ceramide but this finding is preliminary. Importantly, is the requirement on sphingolipid network and ceramide for ZIKV replication really specific? Other lipids should be added as controls. Will ZIKV replication be compromised if other lipid biosynthesis pathways are perturbed? Will NS4B colocalized with other lipids? There are insufficient controls and pathway analysis done to ascertain their claims!

We thank the reviewer for the opportunity to expand upon the central claim of the paper, which is that the presence of ceramides in replication site membranes – facilitated by virus-induced changes in sphingolipid metabolism – is required for ZIKV replication. As they point out, this raises the question of whether this mechanism is specific for ceramide, or whether depletion of any of the various lipid species present in replication-site membranes would similarly block viral replication. To address this question, we took advantage of a panel of lipid biosensors used by us in previous studies [PMID: 29549788] to investigate the metabolism and localization of the phosphatidylinositol (PI) species phosphatidylinositol-4-phosphate (PI4P) and phosphatidylinositol 4,5-bisphosphate [PI(4,5)P₂] during ZIKV infection. PI4P is a lipid that had previously been shown by Hsu et al. [PMID:20510927] to be enriched in the replication-site membranes of Hepatitis C virus (HCV) and other positive-strand RNA viruses. Indeed, we observed

colocalization between a fluorescent biosensor for PI4P and NS4B, indicating the presence of this lipid at the ZIKV replication complex (Supplementary Fig. 10b). However, PI(4,5)P₂, an interconvertible PI species known to be involved in cellular signaling [PMID:30154550], did not colocalize with NS4B (Supplementary Fig. 10a).

To determine the specificity of sphingolipids in ZIKV infection, we sought to examine the roles of other lipids such as PIs and cholesterol in ZIKV replication. Pharmacological inhibition of the principal PI4P-producing enzyme PI4P-kinase (PI4KIII) with the small molecule inhibitor PIK93 did not significantly affect ZIKV replication (Supplementary Fig. 6), in contrast to similar experiments by Hsu et al. showing that inhibition of PI4P biosynthesis blocked HCV and enterovirus replication. On the other hand, treating cells with lovastatin, an inhibitor of cholesterol biosynthesis, resulted in a more modest (~30 %) but significant reduction in ZIKV replication at 24 h post-infection. This result is as expected and recapitulates findings with other flaviviruses by Mackenzie et al. [PMID: 18005741]. In comparison, inhibiting sphingolipid biosynthesis with myriocin or FB1 treatment resulted in an order-of-magnitude reduction in viral shedding at 24 hours after infection (Fig 3b). Moreover, ZIKV infection in sphingomyelin synthase-1 mutant cells resulted in about 100-fold more virus shedding as compared to control cells (Fig 6a). These data show that the role of ceramide and the sphingolipid pathway in ZIKV replication is specific, and not a general bulk effect. We have described these data in the text (Lines 204-211).

3. Since the ceramide supplement experiment was not successful, other experiments must be done to demonstrate the physiological importance of the sphingolipid network as also suggested by the authors. This MUST be performed with KO mice and/or inhibitors in vivo.

We thank the reviewer for their emphasis on physiologically relevant results, and in our experiments with human neural progenitors and neuroblastoma cells (Fig. 3g, h) have demonstrated that the sphingolipid network is required for ZIKV replication in its physiological host cell environment.

4. In cell-based lipidomics or metabolomics, the cell number is essential and would significantly affect the MS data analysis. The cell counting, DNA and protein quantification in different groups should be used to normalize the MS data for appropriate statistical analyses. Although the authors indicated that no significant cytopathic effects were observed and that the measured protein showed no differences, the protein quantification results should be provided to unambiguously show the lack of differences between the mock-infected and virus-infected groups.

While we had this control in our first submission, we did not include the protein content graph in the supplementary information of our resubmission. That graph has been placed in Supplementary Fig. 1a, where it confirms that there are no significant differences in the number of cells used for lipidomics.

5. Lipids standards / internal controls should be shown for the lipid extraction to assess the variance of the individual lipid species in different samples' extractions, the extracted lipid efficiency and coverage for the label-free relative quantification approach in global LC- MS-based lipidomics experiments.

We thank the reviewer for their comment, which was also raised by Reviewer #3 in the initial submission. In Supplementary Fig. 1f-g we have added lipid abundance boxplots before normalization, which along with the existing post-normalization boxplots, confirm that there are

no notable differences between the extracted lipid samples in our analysis. To further determine if there is any significant variation between samples, we performed an additional statistical test. Briefly, the variance of abundances across lipid species for an individual sample was compared to the variance of abundances for a different sample; this was done for all pairwise sample comparisons in a dataset. Further, data from both ionization methods and the datasets before and after normalization were run. More specifically, the null hypothesis $H_0: \sigma_i^2 = \sigma_j^2$ for all samples i and j , such that $i \neq j$ within each dataset. An F-test for equal variances was conducted to test the null hypothesis against the alternative hypothesis that variances were not equal. Across all datasets, both before and after normalization, p-values ranged from 0.1219 to 0.6966, indicating that there is no evidence of a difference in variability across lipid species from sample to sample. For the underlying data, please see the Source Data file for Fig. 1, where the relative intensities for individual lipid species in each biological sample are shown.

As we indicated in our previous resubmission, our global lipidomic analysis was performed using Orbitrap mass spectrometer that is based on exact mass/fragmentation identification. Internal standards were not used for this study. To facilitate quantification of lipids, a reference database for lipids identified from the MS/MS data was created and features from each analysis were then aligned to the reference database based on their identification, m/z and retention time using MZmine 2. Aligned features were manually verified and peak apex intensity values were exported for subsequent statistical analysis.

The approach used in this manuscript, label-free quantitation, has been employed for the last two decades in LC-MS-based proteomics and is accurate in identifying relative abundance differences in detected peptides. Therefore, the same approach is reasonable in global LC-MS-based lipidomics experiments. Indeed we and others have utilized the label-free relative quantification approach in several peer-reviewed publications and shown it to be successful at identifying relative differences in lipid species in global LC-MS-based lipidomics experiments (Dautel et al., 2017; Diamond et al., 2010; Eisfeld et al., 2017; Kyle et al., 2018, 2019; Perera et al., 2012; Sorensen et al., 2010; t'Kindt et al., 2015; Telenga et al., 2014; Tisoncik-Go et al., 2016). We have made sure the Methods section of the manuscript describes label-free relative quantification.

Reviewer #3 (Remarks to the Author):

The authors have addressed my questions and comments sufficiently. Thank you. Ekin Atilla-Gokcumen

Thank you, Dr. Atilla-Gokcumen. We are very pleased that we were able to address all of your comments satisfactorily, especially given your expertise in mass spectrometry-based lipidomics.

Reviewer #4 (Remarks to the Author):

I have gone through the report of referee 1 and the responses to this referee. I find the responses satisfactory. I did not go into the details of the lipidomics data pipeline because this was also addressed by referee 3 who will also get a chance to look at the revision. I do not understand why the authors chose to ignore the changes in PC and PI (even PC/PE ratios could be important). When they analyse their data they look mainly at fold changes and therefore, the changes in ceramide look stronger than for PC and PI, but PC and PI are more abundant lipids, so it is more difficult to get large fold changes.

We thank the reviewer for their time spent reviewing the comments of the first referee. We are pleased that they are satisfied with our responses. In response to their own comment, we chose to focus our investigation on sphingolipids, and ceramides in particular, due to the large magnitude of their changes in ZIKV-infected cells and known roles in important cellular processes. In preparing our revised manuscript, we observed a consistent pattern of ceramide regulation in NS4B-transfected cells, giving further weight to these lipids as important factors in ZIKV replication. However, we do not doubt that there are changes in other lipid classes, such as PC and PI, that are potentially relevant to infection and deserve further study in the future. We have carried out preliminary investigations into other lipids such as cholesterol and some PI (see Supplementary Fig. 6 and 10), where results indicated a potential role for these lipids in ZIKV replication, though to a much lesser extent than ceramide.

Reviewers' Comments:

Reviewer #4:

Remarks to the Author:

In my opinion the authors have responded well to the reviewers comments. There are many ways to quantify lipids. I agreed that for relative quantitation their label-free method without internal standards should be fine. Of course, they cannot make conclusions about absolute amounts, but this is not the goal here. They have added additional experiments further supporting the role of ceramide and suggesting that some other lipids might also have effects, albeit less. While these experiments do not provide evidence for the mechanism of ceramide action, they should be sufficient for this publication. Examining mechanism should be the goal of future work.

Response to referees' comments

Reviewer #4 (Remarks to the Author):

In my opinion the authors have responded well to the reviewers comments. There are many ways to quantify lipids. I agreed that for relative quantitation their label-free method without internal standards should be fine. Of course, they cannot make conclusions about absolute amounts, but this is not the goal here. They have added additional experiments further supporting the role of ceramide and suggesting that some other lipids might also have effects, albeit less. While these experiments do not provide evidence for the mechanism of ceramide action, they should be sufficient for this publication. Examining mechanism should be the goal of future work.

We thank the referee for stepping in to review our revised manuscript. We are currently pursuing more mechanistic studies into the role of ceramide in viral replication.